# HUMAN-INSPIRED EPISODIC MEMORY FOR INFINITE CONTEXT LLMs

**Zafeirios Fountas**[1], **Martin A Benfeghoul**[1,*], **Adnan Oomerjee**[1,*], **Fenia Christopoulou**[1],
**Gerasimos Lampouras**[1], **Haitham Bou-Ammar**[1,2] **and Jun Wang**[2]

[1]Huawei Noah's Ark Lab, London, UK

[2]AI Centre, Department of Computer Science, University College London, London, UK
`{zafeirios.fountas,adnan.ebrahim.oomerjee}@huawei.com`
`{gerasimos.lampouras,haitham.ammar}@huawei.com`
`martin.antoine.benfeghoul@h-partners.com`
`jun.wang@ucl.ac.uk`

## ABSTRACT

Large language models (LLMs) have shown remarkable capabilities, but still struggle with processing extensive contexts, limiting their ability to maintain coherence and accuracy over long sequences. In contrast, the human brain excels at organising and retrieving episodic experiences across vast temporal scales, spanning a lifetime. In this work, we introduce EM-LLM, a novel approach that integrates key aspects of human episodic memory and event cognition into LLMs with no fine-tuning, enabling them to handle practically infinite context lengths while maintaining computational efficiency. EM-LLM organises sequences of tokens into coherent episodic events using a combination of Bayesian surprise and graph-theoretic boundary refinement in an online fashion. When needed, these events are retrieved through a two-stage memory process, combining similarity-based and temporally contiguous retrieval for efficient, human-inspired access to relevant information. Experiments on the LongBench and ∞-Bench benchmarks demonstrate EM-LLM's superior performance, consistently outperforming the state-of-the-art retrieval model InfLLM across various baseline LLMs. In addition, EM-LLM outperforms its popular counterpart, RAG, in a wide range of tasks, while requiring similar resources. Notably, EM-LLM's performance even surpasses full-context models in most tasks, while successfully performing retrieval across 10 million tokens – a scale computationally infeasible for such models. Finally, our analysis reveals strong correlations between EM-LLM's event segmentation and human-perceived events, suggesting parallels between this artificial system and its biological counterpart, thereby offering a novel computational framework for exploring human memory mechanisms.

## 1 INTRODUCTION

For contemporary pre-trained large language models (LLMs), the context window serves as the primary mechanism to incorporate domain-specific, private, or common up-to-date information. However, despite their remarkable and ever-expanding capabilities, LLMs still exhibit significant limitations when tasked with processing extensive contexts (Liu et al., 2024a). These limitations stem from inherent challenges in Transformer-based architectures. Recent studies have shown that Transformers struggle with extrapolating to contexts longer than their training window size (Kazemnejad et al., 2024). On top of this, employing softmax attention over extended token sequences requires substantial computational resources for each token generation, while the resulting aggregated embeddings (the weighted sums of value vectors) risk becoming excessively noisy and losing their distinctiveness (Tworkowski et al., 2023).

To mitigate these challenges, recent works have focused on retrieval-based methods, either in the form of in-context augmentation (e.g., retrieval-augmented generation (RAG)-based techniques (Lewis et al., 2020; Gao et al., 2024)) or via retrieval of previously-inferred key-value pairs (KV) within

---

*Equal Contribution. Code available at: `https://github.com/em-llm/EM-LLM-model`

individual attention heads (Wu et al., 2022; Tworkowski et al., 2023; Bertsch et al., 2023). Notably, state-of-the-art (SOTA) performance is achieved when KV pairs are initially organised into non-overlapping segments and then retrieved together as one block of sequential tokens (Xiao et al., 2024a). While such techniques present interesting research avenues, we still see a significant gap between the performance of LLMs in short- vs long-context tasks, even when existing long-context architectures are employed (Liu et al., 2024a).

This work tackles the above challenges and attempts to bridge this performance gap by taking inspiration from the algorithmic interpretation of *episodic memory in the human brain* – the memory system responsible for encoding, storing, and retrieving personal experiences and events. The brain makes sense of its continuous experience in the real world by segmenting it into discrete episodic events (Clewett et al., 2019; Zacks, 2020), which are first organised in a hierarchical and nested-timescale structure (Baldassano et al., 2017) and then stored in long-term memory. Notably, the boundaries between such events are the access points for memory retrieval (Michelmann et al., 2023a) and are widely believed to correspond to points in time with high prediction errors between the brain's generative model and its raw sensory input (a.k.a., *surprise*). In this context, surprise refers to moments when the brain's predictions about incoming sensory information are significantly violated, leading to a mismatch between what is expected and what is actually perceived. These instances of high surprise are thought to signal important changes in the environment or narrative, prompting the brain to segment the ongoing experience into distinct events (Zacks et al., 2007; 2011; Roseboom et al., 2019; Sinclair et al., 2021; Fountas et al., 2022). Once segmented and stored, the brain recalls episodic memories based on their similarity to current experience, recency, original temporal order, and their proximity to other recalled memories (temporal asymmetry and contiguity (Howard and Kahana, 2002)).

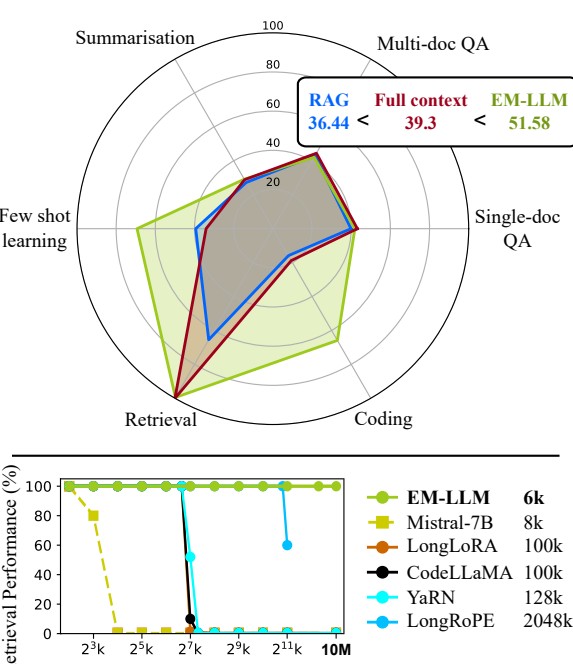

Figure 1: **Top:** EM-LLM$_S$ (surprise only) vs. RAG (NV-Embed-v2 retriever) vs. full-context, with LLaMA-3.1-8B as the base LLM, evaluated on LongBench. **Bottom:** Comparison of various long-sequence methods (sorted based on their context window length) on an extended version of $\infty$-Bench's *Retrieve.PassKey*. Baseline data taken from Ding et al. (2024).

**Contributions:** We propose *EM-LLM*, a novel architecture integrating crucial aspects of event cognition and episodic memory into Transformer-based LLMs through three key innovations (a, b and c). For memory formation, we segment input token sequences into memory units representing episodic events. The boundaries of these units are (a) initially determined using the model's surprise level during inference, then (b) refined to maximize within-unit cohesion and cross-unit separation (see Section 3.2). This refinement leverages graph-theoretic metrics, treating attention key similarity as a weighted adjacency matrix, and aims to enhance efficient information recall in complex, long-context tasks: by consolidating related information into single units, we seek to minimize the number of units needed for event-specific recall. The resulting memory formation process is computationally efficient: surprise-based segmentation requires no additional computation, and refinement complexity is $\mathcal{O}(nm)$, where $m$ is typically negligible compared to the token count $n$ in long-context tasks. For memory recall, (c) our approach combines similarity-based retrieval with temporal contiguity and asymmetry mechanisms, building on recently discovered parallels between LLMs and human sequential information retrieval patterns (Ji-An et al., 2024). This method therefore ensures efficient information access while replicating temporal dynamics from human free recall studies (Howard and

Kahana, 2002), and enhancing performance on tasks requiring temporal reasoning. See Appendix E.2 for analysis of EM-LLM's architectural contributions.

**Performance:** We show that our method is scalable and significantly outperforms the SOTA retrieval model InfLLM (Xiao et al., 2024a), as well as RAG and full-context methods, on the widely-used LongBench (Bai et al., 2023) and ∞-Bench (Zhang et al., 2024) benchmarks designed for long-context tasks (see Fig. 1). Furthermore, we perform successful passkey retrieval across 10M tokens, a length which is computationally infeasible for current full-context models. To further prove our hypotheses, we then employ a series of human-annotated podcast scripts to show that information in LLM attention heads can be semantically grouped in a way that correlates with the event structure perceived by humans. Therefore, LLM-perceived surprise can indeed serve as a proxy for the cognitive signals that drive human event segmentation, as confirmed by previous studies (Kumar et al., 2023). Finally, using the long-context PG-19 dataset (Rae et al., 2020), which comprises a diverse corpus of English books, we evaluate the effectiveness of our segmentation method for grouping relevant information and assess the performance of different boundary refinement objectives.

## 2 RELATED WORK

### 2.1 LONG-CONTEXT IN LLMS

Recently, several approaches have been proposed to extend the context window of Transformer-based models. These include methods that address the limited representational capacity of softmax attention, and its quadratic computational and memory cost (Katharopoulos et al., 2020; Munkhdalai et al., 2024). Other methods target the poor extrapolation of typical positional encodings to out-of-distribution context lengths (Kazemnejad et al., 2024). The latter is evident in most widely used methods, including the original absolute positional encodings (Vaswani et al., 2017) and the more recent relative positional encodings, such as the Rotary Positional Embeddings (RoPE) (Su et al., 2024). To address this, some methods propose scaling of the rotation angles (Chen et al., 2024a) or the base constant in RoPE (Xiong et al., 2023; Liu et al., 2024b; Peng et al., 2024; Ding et al., 2024). Others, scale positions without affecting the embedding function (Press et al., 2021; Chen et al., 2023; Jin et al., 2024), explore alternative strategies such as KERPLE (Chi et al., 2022) and FIRE (Li et al., 2024a) or adopt relative position mechanisms from certain LMs like T5 (Raffel et al., 2020).

Concerning computational efficiency and diluted attention, successful approaches propose methods for general improvements to Transformer efficiency through optimised computations (Dao, 2024; Han et al., 2024a; Aminabadi et al., 2022; Kwon et al., 2023; Liu et al., 2024c; Brandon et al., 2023) or compression techniques (Nawrot et al., 2024; Zhang et al., 2023), as well as training methods tailored for long-context scenarios (Zhu et al., 2024; Chen et al., 2024b). Another direction is the utilisation of retrieval-based methods, the vast majority of which relies on a vector database that keeps a key-value cache and scalable approximations of k-nearest neighbors (k-NNs) to perform lookups (Wu et al., 2022; Tworkowski et al., 2023; Bertsch et al., 2023). Interestingly, since using a key-value cache with k-NN lookup can be seen as an approximation of applying softmax attention to the full token sequence (see Appendix F.1), k-NN retrieval methods can be used without fine-tuning (Bertsch et al., 2023). For an exception that does not rely on k-NNs, see Wang et al. (2023).

A recent and interesting variant of k-NN retrieval involves retrieving large groups of tokens, rather than individual ones. Models that rely on this approach include SLED (Ivgi et al., 2023) and the more recent InfLLM (Xiao et al., 2024a), which achieves SOTA performance on long-context benchmarks. InfLLM segments the entire context length into fixed-size memory units and employs k-NN lookup using the tokens with the highest accumulated scores per unit. The latter can be seen as a form of hierarchical attention in models that use such retrieval, as illustrated in Fig. 2. While group-based retrieval represents a promising direction, our approach significantly advances this concept by dynamically determining token groupings in a manner akin to human memory formation. This effectively addresses a fundamental limitation of InfLLM's fixed-size segmentation and enables more adaptive and context-sensitive processing of extended information.

### 2.2 NEURAL MODELS OF EPISODIC MEMORY AND EVENT COGNITION

The concept of episodic memory, central to our approach, has been extensively studied in both theoretical neuroscience and machine learning. Neural models of episodic memory capture human behaviour and neuroimaging data, providing insights into how the brain processes and stores experiences and suggesting links between memory, efficient representations and navigation of physical

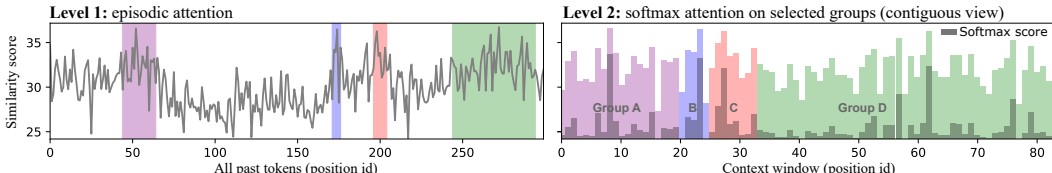

Figure 2: Group-based $k$-NN retrieval can be seen as a form of hierarchical episodic attention. Initially, $k = 4$ groups of tokens are selected (left) and then used for softmax attention (right), as if all other similarity scores were forced to be zero (non-shaded areas of the left curve). This framework can support multiple levels of episodic attention.

and conceptual spaces (Gershman et al., 2012; Benna and Fusi, 2021). In machine learning, episodic memory-inspired approaches have yielded significant improvements across various domains. For instance, episodic control has enhanced reinforcement learning agents' performance and learning speed (Blundell et al., 2016; Pritzel et al., 2017; Coda-Forno et al., 2024). In addition, models of memory construction and consolidation have been successful in alleviating catastrophic forgetting in neural networks (Kirkpatrick et al., 2017; Lopez-Paz and Ranzato, 2017; Chaudhry et al., 2019; Buzzega et al., 2020; Prabhu et al., 2020), including LLMs (Das et al., 2024), and appear to explain key features of human memory, such as imagination and future thinking (Spens and Burgess, 2024).

These models have revealed key aspects of episodic memory, particularly in describing how experiences are segmented into events, and when new memories are encoded and retrieved (Lu et al., 2022). Surprise plays a critical role in this process, triggering event boundaries and memory formation (Fountas et al., 2022; Kumar et al., 2023). This event-based structure is deeply intertwined with our perception of time (Roseboom et al., 2019; Sherman et al., 2022), highlighting the interdependence of memory and temporal cognition. This insight has helped generative models for video (Zakharov et al., 2022a;b) and reinforcement learning (Zakharov et al., 2021) to capture temporal dynamics more accurately. In terms of memory retrieval, studies in human free recall have shown a distinctive increased likelihood of retrieving items encoded close together in time (temporal contiguity) and in succession (temporal asymmetry) (see Fig. 3A). Recently, it was shown that attention heads in Transformer-based LLMs that are associated with in-context learning, already exhibit the same dynamic retrieval behaviour (Ji-An et al., 2024) (Fig. 3B) including both contiguity and asymmetry effects. Therefore, Transformers have the inherent ability to act as episodic memory retrieval models, if provided with the right information within their context window. Our work leverages these concepts of surprise-based event segmentation and LLMs' inherent temporal contiguity and asymmetry effects to enable a new generation of Infinite Context-Length LLMs, capable of processing and understanding information over vastly extended timescales.

## 3  EM-LLM: LLM WITH EPISODIC MEMORY

### 3.1  ARCHITECTURE

EM-LLM is designed to be applied directly to pre-trained LLMs, enabling them to handle context lengths significantly larger than their original training length. Our architecture, illustrated in Fig. 3C, divides the context into three distinct groups: initial tokens, evicted tokens and local context. This structure, while incorporating insights from recent work on token block retrieval (Xiao et al., 2024a), introduces novel elements inspired by human episodic memory.

The local context represents the most recent tokens, maximising information about the current task, and fits within the typical context window of the underlying LLM. This group utilises full softmax attention and plays a role similar to the focus of attention in cognitive models of working memory, holding the most immediately relevant information for the current task (Cowan, 2001). The evicted tokens typically comprise the majority of past tokens in a long-context scenario, extending far beyond the LLM's original training length. These tokens are managed by our proposed memory model functioning similarly to short-term episodic memory in the brain. Finally, following previous work, we also maintain a group of 128 initial tokens in the LLM context. These act as *attention sinks* and help recover the performance of window attention, as first observed by Xiao et al. (2024b); Han et al. (2024b) and later adopted by Xiao et al. (2024a). For retrieved tokens, which are therefore discontinuous and outside the local context, we assign a fixed position embedding as in Raffel

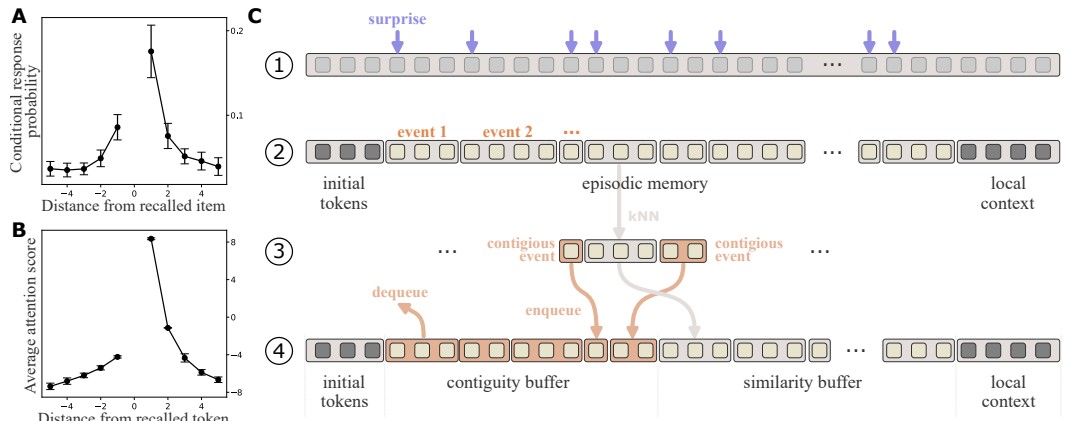

Figure 3: (**A**) Example of the temporal contiguity and asymmetry effect in human free recall. Data averaged over several large free recall studies (adopted from Howard and Kahana (2002)). (**B**) The attention scores of a GPT2 head averaged over all tokens tested (adopted from Ji-An et al. (2024)). (**C**) Schematic illustrating our proposed process for memory formation and retrieval in each layer: ① Input sequence with surprise-based segmentation (purple arrows indicate high surprise). ② Formation of episodic memories: input is segmented into events and stored, with initial tokens and local context preserved. Note that the boundary refinement process is not shown here for clarity. ③ Memory retrieval via k-NN search, selecting contiguous events from episodic memory. ④ Final context window structure, comprising initial tokens, contiguity buffer (populated by neighbouring events), similarity buffer (from k-NN retrieval), and local context.

et al. (2020); Xiao et al. (2024a). This architecture enables EM-LLM to effectively process and utilise information from positions outside its pre-trained local context window, while maintaining the underlying LLM's performance characteristics.

## 3.2 MEMORY FORMATION VIA SURPRISE

In the context of LLMs, we define episodic memory as the organised, event-based collection of past key-value pairs, analogous to the latent representations of personal experiences in human memory. Just as unexpected or novel information plays a crucial role in human memory formation, we posit that analogous indicators of novelty in LLMs can serve as an effective proxy for identifying significant "events" within the model's experience. In Bayesian terms, surprise is quantified by the negative log-likelihood of observing the current, ground-truth token given the previous tokens in an auto-regressive model, with high values indicating the unpredictability or novelty of each new token within the context according to the model, i.e., being "surprised" by the next token. Following work on cognitive modelling (Roseboom et al., 2019; Fountas et al., 2022), we employ a thresholding mechanism to perform an initial identification of event boundaries (used for the first time in LLMs). Formally, a token $x_t$ is considered a potential boundary if its surprise value exceeds a threshold $T$:

$$-\log P(x_t|x_1, \ldots, x_{t-1}; \theta) > T \qquad \text{with} \qquad T = \mu_{t-\tau:t} + \gamma\sigma_{t-\tau:t} \qquad (1)$$

where $\mu_{t-\tau:t}$ and $\sigma^2_{t-\tau:t}$ are the mean and variance of surprise for a window offset $\tau$, and $\gamma$ is a scaling factor. The choice of threshold $T$ is critical in balancing the granularity of segmentation with the model's sensitivity to contextual shifts. If the $T$ is too high, we will identify very few event boundaries, especially if the local context contains few surprising tokens. Conversely, a low $T$ results in frequent boundary identification. Using a moving window ensures that $T$ adapts to contextual shifts, minimizing the need for manual tuning while maintaining control over threshold sensitivity via $\gamma$. This initial segmentation results in a set of potential event boundaries $\mathcal{B} = b_1, b_2, ..., b_k$, where each $b_i$ represents the index of a token exceeding the surprise threshold. These boundaries serve as the starting point for our subsequent refinement process, which aims to optimise the intra-event coherence and inter-event distinctiveness of the resulting memory segments.

## 3.3 BOUNDARY REFINEMENT

While surprise-based segmentation provides an effective initial estimate of event boundaries, we make the key observation that the utility of elements within an event, during memory recall, depends

---

**Algorithm 1** Event segmentation in KV cache

---

**Input:** tok: List of tokens in the sequence
**Input:** $T$: Threshold for surprisal to identify initial boundaries
**Input:** $f$: Metric function to evaluate potential boundaries
**Output:** $\mathcal{B}$: List of final boundary positions
1: $\mathcal{B} \leftarrow$ [i for i in range(`length(tok)`) if $-\log(P(\texttt{tok}[i])) > T$]    ▷ Boundary identification
2: **for** i in range(length($\mathcal{B}$)) **do**
3:     $\alpha, \beta = \mathcal{B}[i], \mathcal{B}[i+1]$
4:     $\mathcal{B}[i+1] \leftarrow \arg\max_{\hat{\beta} \in (\alpha, \beta]} f(A, \{\alpha, \hat{\beta}\})$    ▷ Boundary refinement
5: **end for**
6: **return** $\mathcal{B}$

---

on their likelihood of being utilised by the current query. Therefore, we theorise that memory recall will be most efficient with high intra-event similarity between keys while maintaining low inter-event similarity. For instance, see the similarity of groups in Fig. 2. To further ensure this, we introduce a boundary refinement step that looks to optimise this objective. Such an objective is typically optimised in the context of graph-clustering, hence we express this refinement process in a graph-theoretic manner. To achieve this, we treat the similarity matrix between all keys of an attention head $h$ within the local context window for tokens $x_1, x_2, ..., x_n$ as an adjacency matrix $A^h$:

$$A_{ij}^h = \text{sim}(K_i^h, K_j^h), \tag{2}$$

where $K_i^h$ and $K_j^h$ are the key vectors corresponding to tokens $x_i$ and $x_j$, respectively. The similarity function measures the closeness of two key vectors; in our implementation, we use dot product similarity $K^h{}_i^\mathsf{T} \cdot K_j^h$ due to its effectiveness in capturing semantic relationships in high-dimensional spaces (Vaswani et al., 2017) and to align with the mechanism of self-attention in Transformers.

To evaluate the quality of potential boundaries, we define a metric function $f(A, \mathcal{B}) : \mathbb{R}^{n \times n} \times \{1, \ldots, n\}^k \to \mathbb{R}$. This function quantifies the cohesion within events and separation between events based on the graph structure represented by the similarity matrix $A$ and event boundaries $\mathcal{B}$. We experiment with two widely-accepted graph-clustering metrics: *modularity* and *conductance* (Miasnikof et al., 2018). Modularity (Newman and Girvan, 2004) provides a measure of the quality of a particular division of a network into communities, with higher values indicating higher edge density in the identified cluster when compared to the density of edges expected in a random cluster. As our edge weights represent the similarity between two tokens, we seek to maximise this metric. Modularity is defined as:

$$f_M(A^h, \mathcal{B}) = \frac{1}{4m} \sum_{i,j} \left[ A_{ij}^h - \frac{1}{2m} \sum_i A_{ij}^h \cdot \sum_j A_{ij}^h \right] \delta(c_i, c_j) \tag{3}$$

where $m$ is the total edge weight in the graph, $c_i$ is the community (episodic event) to which node $i$ is assigned, and $\delta$ is the Kronecker delta function. Conductance, on the other hand, measures the fraction of total weighted edges cut by a given community boundary, and is defined as:

$$f_C(A^h, \mathcal{B}) = \min_{S \in V} \frac{\sum_{i \in S, j \notin S} A_{ij}^h}{\min(\text{vo}(S), \text{vo}(V \setminus S))}, \quad \text{with vo}(S) = \sum_{i,j \in S} A_{ij}, \ \text{vo}(V \setminus S) = \sum_{i,j \notin S} A_{ij} \tag{4}$$

where $S = \{b_i, b_i + 1, ..., b_{i+1}\}$ is a subset of all nodes $V = \{b_1, b_1 + 1, ..., b_k\}$ in the induced graph, with $b_i \in \mathcal{B}$. Lower conductance values indicate better community structure. Our boundary refinement algorithm sequentially adjusts the initial surprise-based boundaries to optimise these metric functions. While our best results are achieved using modularity, we also include comparisons with conductance-based boundary refinement to provide a comprehensive analysis. The overall process is summarized in Algorithm 1 and further discussed in Appendix E.3.

This algorithm first identifies initial boundaries based on the surprise threshold $T$, then refines these boundaries by finding the optimal position $\hat{\beta}$ between each pair of consecutive initial boundaries $(\alpha, \beta)$ that optimises the chosen metric function $f$ (either maximising modularity or minimising conductance). This process ensures that the final segmentation (1) captures points of high surprise and (2) optimises for coherent information grouping. The boundary identification step incurs negligible computational cost, as it only evaluates existing LLM outputs. The time complexity of Algorithm 1 has an overall complexity of $\mathcal{O}(nm)$, where $n$ is the $n$ is the sequence length and $m$ is the chunk size selected to process the sequence (for details see Appendix C.1).

| Base LLM | Method | LongBench | | | | | | | $\infty-$Bench | | | | | |
|---|---|---|---|---|---|---|---|---|---|---|---|---|---|---|
| | | SQA | MQA | Sum | FSL | Ret | Cod | Avg. | C.D | M.F | MC | R.KV | R.P | R.N |
| Mistral v2 | InfLLM (4k+2k) | **33** | 25.5 | 27.1 | 66.1 | 64 | 54.8 | 41.9 | **29.4** | 26.6 | **43.2** | 95.6 | 100 | 99.8 |
| | EM-LLM$_{SM+C}$ | 32.9 | **27** | **27.2** | **66.8** | **84.1** | 54.8 | **43.7** | 28.2 | **27.1** | 42.8 | **99** | 100 | 99.8 |
| LLaMA 3 | InfLLM (4k+4k) | 38.5 | 36.9 | 27 | 69 | 84 | **53.2** | 47 | 30.5 | **23.7** | **43.7** | **5** | 100 | 99 |
| | EM-LLM$_S$ | **39.3** | **37.7** | 27.0 | **69.2** | **87.5** | 50.3 | **47.2** | **31.7** | 16.9 | 40.6 | 4.2 | 100 | **99.5** |
| LLaMA 3.1 | InfLLM (4k+4k) | **41.4** | 40.7 | 29 | 69 | 97 | **64.2** | 51.1 | 22.6 | 33.7 | 46.7 | 81 | 100 | 100 |
| | EM-LLM$_{SM}$ | 41.2 | **41.3** | **29.2** | **69.1** | **98.5** | 64.1 | **51.3** | 22.6 | **34** | **47.6** | **90.2** | 100 | 100 |
| Phi 3 | InfLLM (1k+3k) | 28.4 | 24.9 | 25.6 | 52.9 | 7.5 | 57 | 34.5 | | | | | | |
| | EM-LLM$_S$ | **29.2** | **27.1** | **25.9** | **53.5** | **10** | 57 | **35.4** | | | | | | |
| Phi 3.5 | InfLLM (1k+3k) | 31.7 | 28.5 | 23.9 | **56.3** | 11.5 | **40.3** | 34.2 | | | | | | |
| | EM-LLM$_S$ | **31.8** | **31.9** | **24.5** | 55.5 | **13** | 39.5 | **34.9** | | | | | | |

Table 1: EM-LLM performance on LongBench (grouped tasks) and $\infty$-Bench compared to our baseline InfLLM. **S**: surprise threshold, **SM**: surprise threshold and refinement with modularity, **S+C**: surprise threshold and contiguity buffer, **SM+C**: surprise, refinement and contiguity buffer. Each row indicates the number of local + retrieved tokens (eg. 4k+2k) used for both InfLLM and EM-LLM. See Appendix D.1 for parameter choices and Appendix A.1 for more results and significance testing.

## 3.4 MEMORY RETRIEVAL

When inferring a new token, a number of episodic events are selected and become a part of the (extended) context window of the underlying LLM. Our memory retrieval process employs a two-stage mechanism to select relevant episodic events for the LLM's context window (Fig. 3C). First, we retrieve $k_s$ events using $k$-NN search based on dot product similarity between the current query and representative tokens of each event. These representatives, selected as per Xiao et al. (2024a), are the most influential tokens within each event. For large memory stores, we utilise approximate $k$-NN (Douze et al., 2024) to maintain efficiency. These $k_s$ events, retrieved based on their similarity to the current query, form a part of the LLM's context window that we refer to as the *similarity buffer*.

The second stage of our retrieval process introduces another buffer, which we refer to as the *contiguity buffer*, designed to maintain temporal context. Implemented as a queue of size $k_c$, this buffer promotes temporal relationships in retrieval. When an event is retrieved, we also enqueue its neighboring events (within $\pm n$ positions in the original sequence) into this buffer. This mechanism enables the LLM's "induction" attention heads to exhibit the contiguity and asymmetry effects discussed in Section 2.2. The queue structure allows for a natural decay of temporal context as new events are processed, with older or repeated events being dequeued as new ones are added. In total, $k = k_s + k_c$ events are added to the context window, striking a balance between relevance and temporal relationships in a manner analogous to human episodic memory retrieval. Note that each layer retrieves and attends to these $k$ events individually, allowing it to potentially focus on different parts of the context.

## 4 EXPERIMENTS

### 4.1 PERFORMANCE OF EM-LLM ON LONG-CONTEXT TASKS

**Comparison with KV-retrieval-based LLMs** At the time of writing, InfLLM is considered to be the SOTA KV-retrieval method on long-context benchmarks (LongBench, $\infty$-Bench), as well as being the only method which uses group-based k-NN retrieval in LLMs on such benchmarks. We, therefore, employ this model as our first baseline for comparison with our own methods.

Results on both benchmarks (Table 1) show that our method is able to improve on InfLLM across 5 different base LLMs, 80% of individual task groups of LongBench and on the overall average. Note that the table shows the best single method in terms of overall performance for each ablation (see Appendix A.1 for all ablations in methods). Looking at individual task performance across all ablations in methods, EM-LLM is able to surpass InfLLM in all tasks. Notably, we see an especially large jump in performance in the retrieval (*Passage*, *KV*, *Passkey*, *Number*) and QA (*Narrative*, *Qasper*, *MultiField*, *Hotpot*, *2Wiki* and *Musique*) tasks across all ablations, with up to a 40% and 29.7% improvement over InfLLM respectively. Such tasks require the model to identify and retrieve specific information within the input sequence, a challenging test for the model's ability to accurately recall a wide range of detailed information from a large context concurrently. This substantial

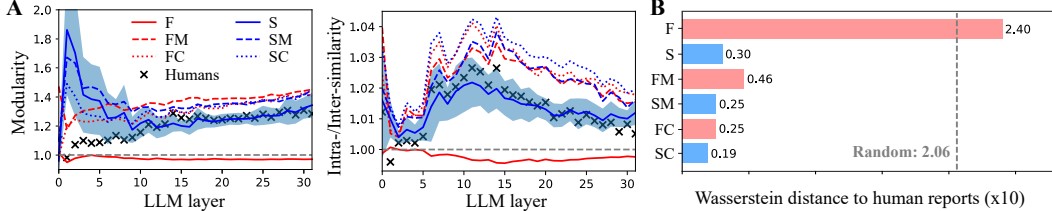

Figure 4: Comparison of human event segmentation with different computational segmentation methods in a human-annotated audio dataset (see also Appendix B). **(A)** Difference in metrics for the cohesion and separation of KV cache of each LLaMA2 layer. The graphs report the difference of each method with the corresponding random segmentation. **(B)** Distance between human reports and different methods. In both sets of results, fixed methods (*F*, *FM*, *FC* | with *M*: Modularity, *C*: Conductance) perform worse than their surprise-based counterparts (*S*, *SM*, *SC*) with InfLLM's method (*F*) performing worse than random.

improvement highlights the effectiveness of our event segmentation method in enhancing long-term memory recall and retrieval accuracy in LLMs.

**Comparison with RAG and full-context LLMs** To evaluate EM-LLM against prominent methods for handling long contexts, we compared its performance on LLaMA 3.1-8B with two different RAG approaches, including the current SOTA NV-Embed-v2 retriever (Lee et al., 2024), as well as with the brute-force baseline of processing all tokens directly within the LLM's softmax attention (full-context). Across most tasks in our benchmarks, EM-LLM outperformed both RAG and full-context methods, as well as a custom surprise-based RAG method (Fig. 1 and Appendix A.2), exceeding the performance of NV-Embed-v2 by $30.5\%$ on LongBench and by $11.5\%$ on $\infty$-Bench.

This significant performance boost over RAG can be attributed to EM-LLM's ability to retrieve and incorporate relevant information at each layer individually, rather than relying on a single retrieval step as in RAG (for an illustration, see Supp. Fig. 5). By accessing more specific and contextually relevant information through layer-wise key-value retrieval, EM-LLM effectively addresses RAG's limitations in precision and lower overall performance (Li et al., 2024b). Additionally, EM-LLM's hierarchical attention avoids the issue of diluted attention in large context windows that affects full-context models, enabling it to outperform both RAG and full-context LLMs on the LongBench dataset. Furthermore, EM-LLM demonstrated remarkable scalability by achieving $100\%$ accuracy on the *Passkey.Retrieval* task with sequences up to 10.2M tokens, far beyond the practical limits of full-context LLMs. This highlights EM-LLM's efficiency in handling extremely long contexts, positioning it as a powerful alternative for long-context processing.

### 4.2 HUMAN AND LLM SURPRISE CLUSTER SIMILAR TOKENS TOGETHER

As mentioned in Section 3.2, we employ modularity and conductance as two refinement objectives in our boundary refinement algorithm, due to their qualities in assessing the intra- and inter-event similarities between individual tokens. We will now use such metrics to compare various event segmentation methods, including human event segmentation data. Additionally, we introduce one further, simple metric for this experiment: the ratio between intra- and inter-community similarity (I/IS), calculated for each head and community $S$ as follows:

$$\text{intra} = \sum_{i \in S, j \in S} A_{ij}, \qquad \text{inter} = \sum_{i \in S, j \notin S} A_{ij}, \qquad \text{I/IS} \equiv \frac{\text{intra}}{\text{inter}} \qquad (5)$$

Kumar et al. (2023) found strong correlations between human-perceived events and prediction errors across 3 short podcasts (7-30 minutes), when processing the corresponding transcript with an LLM. Taking advantage of such human-annotated data and results from previous studies on this dataset (Michelmann et al., 2021; Lositsky et al., 2016), we compare the segmentation quality and correlation with human segmentation for each of our methods (Fig. 4) using our similarity metrics.

As shown in Fig. 4A, human-perceived events achieve significantly higher scores in similarity metrics compared to fixed or random events, suggesting that surprise is indeed an important factor for humans in their own perception of events. Furthermore, surprise-only segmentation (**S**) achieves very similar results to humans, while the addition of our refinement algorithm (**SM**, **SC**, **FM**, **FC**) significantly

| LLM | Metric | F | FM | FC | S | SM | SC |
|---|---|---|---|---|---|---|---|
| Mistral-7B | Mod ↑ | -2.3 ± 4.1 | 29.2 ± 44.0 | 6.7 ± 25.9 | 18.6 ± 29.6 | **39.9 ± 55.5** | 29.5 ± 42.7 |
| | Con ↓ | 9.1 ± 8.7 | -16.9 ± 6.7 | -12.5 ± 9.6 | -23.6 ± 9.4 | -24.6 ± 9.3 | **-27.6 ± 9.8** |
| | I/IS ↑ | -4.3 ± 4.0 | 31.2 ± 21.4 | 3.7 ± 14.9 | 17.9 ± 17.0 | **35.3 ± 27.7** | 21.6 ± 22.4 |
| LLaMA2-7B | Mod ↑ | -1.1 ± 4.3 | 13.4 ± 19.5 | 0.6 ± 7.3 | 8.7 ± 16.0 | **18.7 ± 26.4** | 11.5 ± 19.4 |
| | Con ↓ | 11.9 ± 9.8 | -18.8 ± 7.4 | -13.7 ± 10.9 | -29.5 ± 10.2 | -29.7 ± 10.1 | **-33.3 ± 10.3** |
| | I/IS ↑ | -3.8 ± 3.7 | 20.7 ± 184.7 | -1.1 ± 6.8 | 15.0 ± 880.0 | **25.0 ± 19.9** | 16.5 ± 15.4 |
| LLaMA3-8B | Mod ↑ | -1.6 ± 3.6 | 18.9 ± 25.6 | 0.9 ± 11.8 | 13.1 ± 21.5 | **27.0 ± 35.6** | 18.3 ± 28.5 |
| | Con ↓ | 11.3 ± 9.5 | -20.3 ± 6.9 | -14.6 ± 11.4 | -29.7 ± 9.2 | -30.6 ± 9.2 | **-33.9 ± 9.6** |
| | I/IS ↑ | -3.8 ± 3.1 | 24.5 ± 13.9 | -1.1 ± 5.8 | 15.7 ± 11.0 | **28.1 ± 16.1** | 16.4 ± 12.2 |

Table 2: Comparison with graph-theoretic metrics in the KV cache of different LLMs and segmentation methods using the PG-19 dataset and $\gamma = 10^{-3}$. Reported values are the difference with random segmentation. Mod: modularity$\times 10^5$, Con: conductance, I/IS: intra/inter-similarity $\times 10^3$.

improves performance. Fig. 4B further shows that surprise-based methods ($\mathbf{S}$, $\mathbf{SM}$, $\mathbf{SC}$), consistently identify event boundaries that are closest to those perceived by humans.

## 4.3 COMPARING SEGMENTATION METHODS

Our experiments on the PG-19 dataset (see Table 2) clearly demonstrate that surprise-based segmentation with refinement ($\mathbf{SM}$, $\mathbf{SC}$) provides the best results in terms of event similarity metrics, regardless of the base LLM used. While the surprise-only method ($\mathbf{S}$) achieves decent results, we observe that refinement is especially adept to improving this performance with regards to our metrics, as it is directly optimising for such an objective. Interestingly however, the fixed-based refinement methods ($\mathbf{FM}$, $\mathbf{FC}$) do not reach the same performance as their surprise-based counterparts, further showing that the initial segmentation with a surprise threshold is crucial to achieving the best possible balance in intra-/inter-similarity with our methods.

## 4.4 SIMILARITY, CONTIGUITY, RECENCY AND TEMPORAL ORDER

As demonstrated in Tables 1 and 2, along with Fig. 4, each of our ablations show various positive improvements on InfLLM. As mentioned in Section 4.3, refinement has a strong positive impact in improving our similarity metrics. This is seen to translate well to model performance in our experiments, with the addition of refinement achieving the best performance in $60\%$ of tasks across LongBench and $\infty$-Bench (see Tables 3-7), as well as agreeing with human data (Fig. 4). The effects of contiguity are also clearly demonstrated, with the addition of our contiguity buffer achieving the best performance in $44\%$ of tasks. Furthermore, these methods are seen to be complementary, often improving on both individual additions.

However, the fact that certain tasks still appear to benefit more from either surprise-only, refinement, or contiguity, is an interesting result. This is likely due to the nature of the tasks and the varying importance of contiguity across these tasks. Where contiguity is not crucial, adding such a buffer to our context window also reduces the size of the similarity buffer, and therefore provides potentially less directly relevant events. This is compatible with our own findings that a contiguity buffer that is as big or smaller than the similarity buffer yields the best results (see Fig. 13), suggesting that the similarity buffer is still the most crucial part of our approach. This is especially the case when combined with refinement, which we expect is due to the improved similarity of refined events, hence further reducing the need for contiguous events.

## 5 DISCUSSION

**Human studies**  Significant correlations have been found between human event segmentation and prediction errors in both LLMs (Kumar et al., 2023) and video models (Fountas et al., 2022; Mariola et al., 2022). Our results add to this growing body of evidence, demonstrating that LLM-based surprise can serve as a proxy for human event segmentation, in multiple levels of hierarchical abstraction, and that the resulting event structure in EM-LLM's attention heads correlates strongly with human-perceived events. This finding suggests a potential, low-level parallels between LLM mechanisms and human cognitive processes (see also Appendix E.1). Furthermore, our model's use of both similarity-based and temporally contiguous retrieval mechanisms parallels human memory retrieval patterns, allowing for the expression of robust phenomena found in human memory research (Howard and Kahana, 2002). The temporal contiguity effect, where items experienced close together in time

are often recalled together, is a robust phenomenon in human memory research (Howard and Kahana, 2002). Further experiments could deepen our understanding of the connections between EM-LLM and human episodic memory. Following Michelmann et al. (2023b), one could test whether the timing of the event boundaries or the degree of modularity per level that our method produces is closer on average to the human consensus, than individual human subjects. Additionally, exploring how different ratios of the contiguity buffer affect the reproduction of human memory biases, and investigating the impact of recency and initial surprise on event recall, could reveal the extent to which EM-LLM exhibits biases found in free recall studies.

Furthermore, EM-LLM's architecture with differentiated context handling (Section 3.1) invites comparisons to cognitive models of human memory beyond episodic. The local context, holding recent and task-relevant information, resembles the limited-capacity working memory system described by Baddeley (2003). Given that EM-LLM's broader context window includes both local context and retrieved memories, it aligns more closely with Ericsson and Kintsch (1995)'s concept of long-term working memory, which allows rapid access to relevant long-term information beyond traditional capacity limits. Alternatively, our architecture parallels Cowan (2001)'s embedded-processes model, where the local context is the "focus of attention", and the full context window represents the activated portion of long-term memory. Future work could explore these analogies further, using EM-LLM as a test-bed for hypotheses about human memory and working memory capacity limits. Inspired by Baddeley's multi-component model, integrating modality-specific buffers into EM-LLM might enhance performance on multi-modal tasks.

**Machine learning**  In refining event boundaries, we utilised modularity and conductance as metrics for evaluating community structure in the similarity graph of attention keys. While effective in our experiments, we acknowledge that numerous other methods for graph clustering and sequence segmentation could potentially be applied (Fortunato, 2010; Yang et al., 2016). Our choice was motivated by their established theoretical foundations and computational efficiency, though comparative studies suggest performance can vary based on network characteristics (Yang et al., 2016). Interestingly, our surprise-based initial boundary detection shares similarities with Bayesian online change-point detection (Adams and MacKay, 2007), suggesting potential avenues for integrating time series analysis techniques into LLM context processing. Future work could explore whether more sophisticated segmentation or clustering algorithms could improve EM-LLM's performance, particularly for extremely long contexts or streaming data scenarios. Such investigations could enhance our model and contribute to understanding how information is structured and processed in LLMs, bridging the gap between traditional sequence analysis and LLM context processing.

Looking ahead, promising directions for future research include extending our segmentation processes to operate at each layer of the Transformer independently. This could lead to more nuanced and hierarchical representations of episodic memories, following the underlying semantic structure of the input more closely. Additionally, exploring how EM-LLM could be utilised to enable imagination and future thinking has great potential for advancing model-based reinforcement learning and continual learning techniques in LLMs. By leveraging its event-based structure to simulate potential future scenarios or recall past experiences in novel contexts, EM-LLM could enhance an LLM's ability to plan, adapt, and learn continuously from new information.

## 6  CONCLUSION

In this work, we introduced EM-LLM, a flexible architecture that integrates key aspects of human episodic memory and event cognition into Transformer-based LLMs. Our approach enables existing LLMs to effectively process vastly extended contexts without the need for pre-training, demonstrating superior performance on long-context tasks compared to the corresponding SOTA. By combining surprise-based event segmentation, graph-theoretic boundary refinement, and a two-stage memory retrieval process, EM-LLM offers a promising path toward virtually infinite context windows. This capability has the potential to revolutionize interactions with LLMs, enabling continuous, personalised exchanges over extended periods and serving as a viable alternative to traditional RAG techniques. Finally, by bridging insights from cognitive science with machine learning, our approach not only enhances the performance of LLMs on long-context tasks but also provides a scalable framework for computational modelling of episodic and event cognition. We hope this study inspires the community to expand research at the intersection of LLMs and human memory.

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

# List of Appendices

## A   ANALYTICAL RESULTS

### A.1   COMPARISON WITH KV-RETRIEVAL-BASED LLMS

While the vast majority of our results in this section show a statistically significant improvement on InfLLM at the benchmark level ($p < 0.05$ using a two-tailed z-test, except LongBench with Phi-3.5 with $p = 0.23$), it should be noted this isn't the case in the majority of individual tasks. However, given the consistency and frequency of improvements across a large number of such tasks, along with the benchmark-level significance of such improvements, we consider the lower, task-level significance to be largely due to the sample size of individual tasks rather than chance, and believe it is still reasonable and justified to claim an overall improvement on InfLLM. Moreover, including individual task results supports transparency and allows for future works to make more granular comparisons and use of such results.

| Task Type | Task | InfLLM | Max Imp. | EM-LLM | | | |
|---|---|---|---|---|---|---|---|
| | | | | S | SM | S+C | SM+C |
| Single-doc QA | NarrativeQA | 22.12 | **1.49%** | 21.77 | 21.13 | 21.10 | **22.45** |
| " | Qasper | 29.33 | **0.17%** | 29.07 | **29.38** | 29.16 | 28.68 |
| " | MultiFieldQA | 47.42 | **1.46%** | **48.11** | 47.39 | 47.72 | 47.62 |
| Multi-doc QA | HotpotQA | 36.56 | **10.15%** | 39.40 | 39.01 | **40.27** | 38.90 |
| " | 2WikiMQA | 22.31 | **5.20%** | 23.46 | 22.75 | **23.47** | 23.46 |
| " | Musique | 17.68 | **6.17%** | 17.97 | 17.82 | 17.98 | **18.77** |
| Summarisation | GovReport | 31.03 | **1.90%** | 31.40 | **31.62** | 31.10 | 31.43 |
| " | QMSum | 23.49 | **2.13%** | **23.99** | 23.20 | 23.48 | 23.47 |
| " | MultiNews | **26.70** | -0.30% | 26.55 | 26.54 | 26.58 | 26.62 |
| Few shot | TREC | 69.00 | **2.90%** | **71.00** | 70.00 | 70.50 | 70.50 |
| " | TriviaQA | 86.67 | **1.10%** | 86.58 | **87.62** | 87.52 | 87.47 |
| " | SAMSum | 42.52 | **0.45%** | **42.71** | 42.13 | 42.34 | 42.48 |
| Retrieval | PassageRetrieval | 64.00 | **32.69%** | 82.67 | 78.92 | **84.92** | 84.08 |
| Code | LCC | 56.67 | **0.64%** | 55.03 | **57.03** | 54.90 | 56.79 |
| " | RepoBench-P | 52.97 | **1.34%** | 50.49 | **53.68** | 51.06 | 52.86 |
| | **Avg. score:** | 41.90 | **4.50%** | 43.35 | 43.22 | 43.47 | **43.71** |
| Code | Code.Debug | 29.44 | **0.88%** | **29.70** | 28.43 | 28.68 | 28.17 |
| Multiple choice | En.MC | **43.23** | -1.02% | 41.48 | 40.61 | 42.79 | 42.79 |
| Retrieval | Math.Find | 26.57 | **5.38%** | **28.00** | 27.43 | 27.71 | 27.14 |
| " | Retrieve.KV | 95.60 | **3.56%** | 92.20 | 97.20 | 97.60 | **99.00** |
| " | Retrieve.PassKey | 100.00 | 0.00% | 100.00 | 100.0 | 100.00 | 100.00 |
| " | Retrieve.Number | 99.83 | 0.00% | 99.83 | 99.83 | 99.83 | 99.83 |
| | **Avg. score:** | 65.78 | **1.47%** | 65.20 | 65.58 | 66.10 | **66.16** |

Table 3: EM-LLM performance on LongBench and $\infty$-Bench (respectively) compared to our baseline, InfLLM, with Mistral-7B-Instruct-v0.2 as the base LLM and 4K+2K context. **S**: surprise threshold, **SM**: surprise threshold + refinement with modularity, **S+C**: surprise threshold + contiguity buffer, **SM+C**: surprise threshold + refinement with modularity + contiguity buffer. **Max Imp.**: Maximum relative improvement over InfLLM across all EM-LLM variants.

| Task | InfLLM | Max Imp. | EM-LLM | | | |
| --- | --- | --- | --- | --- | --- | --- |
| | | | S | SM | S+C | SM+C |
| NarrativeQA | 22.64 | **8.92%** | 24.47 | 22.50 | **24.66** | 23.03 |
| Qasper | 43.70 | **3.25%** | 44.35 | 44.95 | 45.07 | **45.12** |
| MultiFieldQA | 49.03 | **0.47%** | 49.11 | 48.79 | **49.26** | 48.36 |
| HotpotQA | 49.04 | **0.69%** | 48.97 | 49.19 | 48.74 | **49.38** |
| 2WikiMQA | 35.61 | **8.26%** | 38.44 | 38.08 | **38.55** | 38.08 |
| Musique | **26.06** | -1.46% | 25.68 | 25.19 | 24.64 | 23.92 |
| GovReport | 30.76 | **1.11%** | **31.10** | 30.85 | 30.96 | 30.86 |
| QMSum | 22.70 | **0.31%** | 22.63 | **22.77** | 22.62 | 22.58 |
| MultiNews | **27.57** | -0.91% | 27.32 | 27.28 | 27.30 | 27.29 |
| TREC | **73.50** | 0.00% | **73.50** | **73.50** | **73.50** | **73.50** |
| TriviaQA | **90.91** | 0.00% | **90.91** | **90.91** | **90.91** | **90.91** |
| SAMSum | 42.43 | **1.98%** | 43.24 | **43.27** | 42.91 | 42.84 |
| PassageRetrieval | 84.00 | **4.17%** | **87.50** | 86.00 | 86.00 | 85.00 |
| LCC | 59.88 | **0.94%** | 58.49 | **60.44** | 58.55 | 60.41 |
| RepoBench-P | **46.48** | -3.44% | 42.13 | 44.88 | 42.26 | 44.68 |
| **Avg. score:** | 46.95 | **1.62%** | 47.19 | **47.24** | 47.06 | 47.06 |
| Code.Debug | 30.46 | **5.81%** | 31.73 | 30.20 | **32.23** | 31.73 |
| Math.Find | **23.70** | -27.68% | 16.86 | 16.57 | 16.29 | 17.14 |
| Retrieve.KV | 5.00 | **4.00%** | 4.20 | 4.80 | **5.20** | 5.00 |
| En.MC | **43.70** | -3.07% | 40.55 | 42.36 | 39.74 | 40.61 |
| Retrieve.PassKey | 100.00 | 0.00% | 100.00 | 100.00 | 100.00 | 100.00 |
| Retrieve.Number | 99.00 | **0.67%** | 99.49 | 99.49 | **99.66** | 99.49 |
| **Avg. score:** | **50.31** | -3.38% | 48.81 | 48.90 | 48.85 | 49.00 |

Table 4: EM-LLM performance on LongBench and $\infty$-Bench (respectively) compared to our baseline, InfLLM, with LLaMA-3-8B-Instruct as the base LLM and 4K+4K context. Abbreviations as before.

| Task | InfLLM | Max Imp. | EM-LLM | | | |
| --- | --- | --- | --- | --- | --- | --- |
| | | | S | SM | S+C | SM+C |
| NarrativeQA | 26.64 | **2.25%** | 26.05 | 26.05 | **27.24** | 25.98 |
| Qasper | 44.95 | **1.27%** | 44.41 | **45.52** | 44.71 | 45.41 |
| MultiFieldQA | **52.56** | -0.08% | 52.52 | 52.07 | 52.36 | 52.48 |
| HotpotQA | 52.96 | **2.00%** | **54.02** | 52.49 | 53.37 | 53.90 |
| 2WikiMQA | 45.04 | **4.57%** | 45.72 | 45.60 | 46.61 | **47.10** |
| Musique | 23.98 | **7.76%** | 25.37 | **25.84** | 24.60 | 24.73 |
| GovReport | 34.96 | **0.57%** | 35.04 | **35.16** | 34.94 | 34.81 |
| QMSum | 24.36 | **0.94%** | 24.31 | 24.55 | 24.31 | **24.59** |
| MultiNews | 27.78 | **0.14%** | 27.76 | 27.79 | **27.82** | 27.77 |
| TREC | 71.00 | **0.70%** | **71.50** | **71.50** | **71.50** | 71.00 |
| TriviaQA | **92.44** | 0.00% | 92.34 | 92.24 | 92.43 | **92.44** |
| SAMSum | 43.68 | **0.41%** | 43.31 | 43.65 | 43.63 | **43.86** |
| PassageRetrieval | 97.00 | **2.58%** | **99.50** | 98.50 | 98.00 | 97.50 |
| LCC | 65.82 | **2.48%** | **67.45** | 65.69 | 65.74 | 65.74 |
| RepoBench-P | 62.56 | **2.83%** | **64.33** | 62.54 | 62.18 | 61.87 |
| **Avg. score:** | 51.05 | **1.89%** | **51.58** | 51.28 | 51.30 | 51.28 |
| Code.Debug | 22.59 | 0.00% | 22.59 | 22.59 | 22.59 | 22.59 |
| Math.Find | 33.71 | **6.79%** | **36.00** | 34.00 | 35.43 | 27.71 |
| Retrieve.KV | 81.00 | **19.51%** | **96.80** | 90.20 | 95.40 | 95.20 |
| En.MC | 46.72 | **1.88%** | 44.54 | **47.60** | 46.72 | 45.85 |
| Retrieve.PassKey | 100.00 | 0.00% | 100.00 | 100.00 | 100.00 | 100.00 |
| Retrieve.Number | 100.00 | 0.00% | 100.00 | 100.00 | 100.00 | 100.00 |
| **Avg. score:** | 64.00 | **4.70%** | 66.66 | 65.73 | **66.69** | 65.23 |

Table 5: EM-LLM performance on LongBench and $\infty$-Bench (respectively) compared to our baseline, InfLLM, with LLaMA-3.1-8B-Instruct as the base LLM and 4K+4K context. Abbreviations as before.

| Task | InfLLM | Max Imp. | EM-LLM | | | |
| --- | --- | --- | --- | --- | --- | --- |
| | | | S | SM | S+C | SM+C |
| NarrativeQA | 14.82 | **14.04%** | 15.78 | **16.90** | 16.02 | 16.66 |
| Qasper | 28.71 | **5.29%** | 28.25 | 28.46 | 29.10 | **30.23** |
| MultiFieldQA | 41.54 | **5.66%** | 43.48 | **43.89** | 42.57 | 42.57 |
| HotpotQA | 32.64 | **10.45%** | **36.05** | 34.37 | 33.53 | 31.98 |
| 2WikiMQA | 27.08 | **10.16%** | 28.74 | 28.05 | **29.83** | 27.49 |
| Musique | 15.05 | **29.70%** | 16.53 | 16.76 | **19.52** | 16.49 |
| GovReport | 28.96 | **2.97%** | 29.41 | 29.59 | **29.82** | 29.62 |
| QMSum | 21.64 | **3.00%** | **22.29** | 21.90 | 22.06 | 22.07 |
| MultiNews | **26.32** | -0.49% | 26.07 | 26.16 | 25.89 | 26.19 |
| TREC | 67.00 | **3.73%** | 68.50 | **69.50** | 68.50 | 68.00 |
| TriviaQA | 83.71 | **1.73%** | 83.59 | 84.16 | **85.16** | 85.09 |
| SAMSum | 7.83 | **8.30%** | 8.43 | **8.48** | 7.28 | 8.06 |
| PassageRetrieval | 7.50 | **40.00%** | 10.00 | 9.50 | 9.00 | **10.50** |
| LCC | **60.33** | -0.22% | 59.99 | 60.13 | 60.20 | 60.01 |
| RepoBench-P | 53.70 | **0.37%** | **53.90** | 53.59 | 53.44 | 53.27 |
| **Avg. score:** | 34.46 | **8.98%** | 35.40 | 35.43 | **35.46** | 35.22 |

Table 6: EM-LLM performance on LongBench compared to our baseline, InfLLM, with Phi-3-Mini-4K-Instruct as the base LLM and 1K+3K context. Abbreviations as before.

| Task | InfLLM | Max Imp. | EM-LLM | | | |
| --- | --- | --- | --- | --- | --- | --- |
| | | | S | SM | S+C | SM+C |
| NarrativeQA | 17.82 | **8.98%** | **19.42** | 16.63 | 17.55 | 17.12 |
| Qasper | 31.44 | **3.34%** | 32.38 | 31.71 | 31.43 | **32.49** |
| MultiFieldQA | **45.80** | -2.36% | 43.58 | 44.72 | 44.28 | 44.66 |
| HotpotQA | 41.33 | **11.35%** | **46.02** | 44.78 | 44.43 | 44.89 |
| 2WikiMQA | 27.74 | **8.51%** | 29.68 | 28.68 | **30.10** | 29.27 |
| Musique | 16.39 | **21.23%** | **19.87** | 19.26 | 18.70 | 19.41 |
| GovReport | 26.37 | **3.22%** | 27.05 | **27.22** | 26.76 | 27.13 |
| QMSum | 21.19 | **4.29%** | 21.94 | **22.10** | 21.79 | 21.79 |
| MultiNews | 24.23 | **0.70%** | 24.39 | 24.29 | 24.29 | **24.40** |
| TREC | 67.50 | **2.22%** | 67.50 | **69.00** | 67.50 | 68.00 |
| TriviaQA | **84.66** | -0.99% | 83.82 | 83.82 | 83.82 | 83.49 |
| SAMSum | **16.62** | -0.54% | 15.30 | 14.49 | 16.53 | 15.25 |
| PassageRetrieval | 11.50 | **17.39%** | 13.00 | **13.50** | 13.00 | **13.50** |
| LCC | **38.38** | -3.20% | 36.79 | 37.15 | 36.90 | 37.02 |
| RepoBench-P | 42.30 | **1.75%** | 42.16 | **43.04** | 41.91 | 41.65 |
| **Avg. score:** | 34.22 | **5.06%** | **34.86** | 34.69 | 34.60 | 34.67 |

Table 7: EM-LLM performance on LongBench compared to our baseline, InfLLM, with Phi-3.5-mini-Instruct as the base LLM and 1K+3K context. Abbreviations as before.

## A.2 COMPARISON WITH RAG

### EXPERIMENT DETAILS

In our experiments with Retrieval-Augmented Generation (RAG) baselines, we implemented a standard RAG pipeline consisting of a retriever and a downstream LLM. For each example in a benchmark task, the example context is split into chunks of words each and encoded using the retriever's embedding model into a vector database. A similarity lookup into the vector database is used to retrieve the top $k$ most relevant chunks, which are then fed to the downstream LLM alongside the query and task description.

We conducted experiments using two retriever models—NV-Embed-v2 (Lee et al., 2024) and all-mpnet-base-v2 (Reimers, 2022). NV-Embed-v2 is a SOTA LLM-based retriever that uses that, as of September 2024, ranks first on the Massive Text Embedding Benchmark (MTEB) Leaderboard (Muennighoff et al., 2022). It is a fine-tuned Mistral-7Bv0.1 model with an embedding dimension of 4096, trained using contrastive instruction-tuning on both retrieval and non-retrieval datasets. all-mpnet-base-v2 is a smaller 110M parameter model with an embedding size of 768, built on the BERT-base-uncased architecture, trained using contrastive learning on a dataset of over 1 billion sentence pairs. For each embedding model, we ran experiments using LLaMa-3-8B and LLaMa-3.1-8B as the downstream LLM.

For most experiments, we set chunk size to $l = 300$ and $k = 5$, following the protocol of Li et al. (2024c). As a comparative baseline, we also trialled chunking according to surprise, the results of which are shown in Table 9 (RAG-S). In this experiment, the context was first segmented into blocks by the EM-LLM (S) model, each of which then encoded by the retriever's embedding model. Context was dynamically retrieved to fill a buffer of 1500 words, for fair comparison with fixed-size chunking. The RAG-S variant underperformed the fixed-size implementation - we hypothesise that this was due to the retrieved context in RAG-S consisting of highly disjoint information, due to small chunk sizes and high $k$. Incorporating contiguity into the retrieval mechanism may be sufficient to close this performance gap.

### LIMITATIONS OF RAG

RAG requires the use of additional modules alongside the downstream LLM during the generation process, meaning that the quality of the generated output depends on the representational capacity of these modules in addition to the capability of the LLM. For example, the use of a retriever model that has far fewer parameters than the downstream LLM can limit the LLM's ability to generate the most accurate or contextually appropriate responses, as shown by the gap in performance between the two RAG pipelines on LongBench shown in Table 8. In this case, whilst the downstream LLM may be capable of high performance on a given task, the retriever may not be expressive enough to provide the relevant context needed to solve said task. Additional pre/post-retrieval techniques such as query expansion (Jagerman et al. (2023)), knowledge filtering (Shi et al. (2024)) or answer reranking (Majumder et al. (2021)), may help to bridge potential performance bottlenecks, but these involve further increasing the complexity of the generation pipeline. In contrast, EM-LLM outperforms both RAG models whilst only requiring the use of a single LLM across the entire generation stage, removing the issue of performance bottlenecks seen in RAG pipelines. Furthermore, EM-LLM is a general purpose method that can be applied to practically any existing transformer LLM - our method was implemented using a general purpose patch to attention layer modules that provided compatibility with the Huggingface Transformers Library.

| Task | RAG | | EM-LLM$_{SM}$ |
|---|---|---|---|
| | all-mpnet-base-v2 | NV-Embed-v2 | |
| NarrativeQA | 17.31 | 21.39 | **22.50** |
| Qasper | 40.66 | 41.01 | **44.95** |
| MultiFieldQA | 45.16 | **50.47** | 48.79 |
| HotpotQA | 43.32 | **53.64** | 49.19 |
| 2WikiMQA | **41.48** | 40.41 | 38.08 |
| Musique | 26.78 | **31.56** | 25.19 |
| GovReport | 27.87 | 29.26 | **30.85** |
| QMSum | 22.44 | **23.15** | 22.77 |
| MultiNews | 26.04 | **27.48** | 27.28 |
| TREC | 4.50 | 65.00 | **73.50** |
| TriviaQA | 78.98 | 63.75 | **90.91** |
| SAMSum | 9.00 | 32.85 | **43.27** |
| PassageRetrieval | 54.00 | 54.50 | **86.00** |
| LCC | 10.76 | 19.88 | **60.44** |
| RepoBench-P | 13.02 | 34.77 | **44.88** |
| **Avg. score:** | 30.75 | 39.27 | **47.24** |

Table 8: Comparison of RAG with two different retrievers vs. EM-LLM on the LongBench dataset. All methods use LLaMa-3-8B-Instruct for generation.

| Task | RAG-S | RAG | FC | EM-LLM |
|---|---|---|---|---|
| NarrativeQA | 12.10 | 22.54 | **29.14** | 26.05 |
| Qasper | 36.41 | **45.45** | 45.34 | 44.41 |
| MultiFieldQA | 44.08 | 51.67 | **54.98** | 52.52 |
| HotpotQA | 41.56 | **55.93** | 54.01 | 54.02 |
| 2WikiMQA | 28.84 | 42.93 | **45.95** | 45.72 |
| Musique | 19.04 | 30.90 | **33.52** | 25.37 |
| GovReport | 18.12 | 29.91 | 34.49 | **35.04** |
| QMSum | 19.22 | **24.97** | 25.14 | 24.31 |
| MultiNews | 26.21 | 26.77 | 27.00 | **27.76** |
| TREC | 2.5 | 22.50 | 4.50 | **71.50** |
| TriviaQA | 88.26 | 88.11 | 89.07 | **92.34** |
| SAMSum | 8.09 | 7.56 | 8.68 | **43.31** |
| PassageRetrieval | 16.0 | 65.50 | **100.00** | 99.50 |
| LCC | 11.02 | 13.16 | 19.30 | **67.45** |
| RepoBench-P | 17.39 | 18.66 | 18.33 | **64.33** |
| **Avg. score:** | 25.89 | 36.44 | 39.30 | **51.58** |
| Code.Debug | - | **22.59** | 21.70 | 22.59 |
| Math.Find | - | 35.43 | 26.29 | **36.00** |
| Retrieve.KV | - | 31.80 | 92.60 | **96.80** |
| En.MC | - | **64.19** | 58.07 | 44.54 |
| Retrieve.PassKey | - | **100.00** | **100.00** | **100.00** |
| Retrieve.Number | - | 99.83 | 99.32 | **100.00** |
| **Avg. score:** | - | 58.97 | 66.33 | **66.66** |

Table 9: EM-LLM$_S$ (4K+4K) vs. RAG (NV-Embed-v2 retriever) vs. full-context, with LLaMa-3.1-8B as the base LLM, evaluated on LongBench and $\infty$-Bench. The comparison also includes RAG-S, which is the same RAG retriever but with the same surprise-based segmentation used in EM-LLM results.

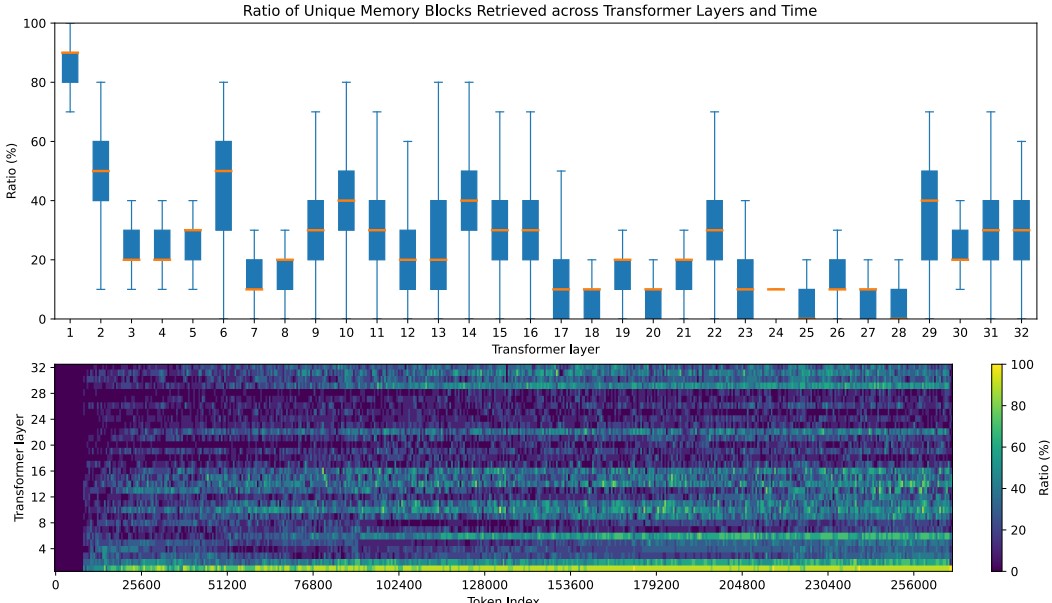

Figure 5: The ratio of blocks retrieved by a layer which were not retrieved by any other layer for the same processed chunk, versus the total number of retrieved blocks by that layer. This is measured using EM-LLM$_S$ with Mistral-7B on a single example of $\infty$-Bench's *Longbook.Choice.Eng* task, with over 500 chunks of 512 tokens. In RAG methods, this ratio would always be zero, as retrieved blocks are used by all layers concurrently.

# B  HUMAN DATA

## B.1  ANALYSIS

The human data released as part of Kumar et al. (2023) used Gaussian smoothing on the average signal across participants to define a probability distribution of likely event boundary positions with respect to timestamps in the podcast. In order to calculate our similarity metrics, as shown in Fig. 4A, we need to express this data in terms of discrete event positions with respect to tokens in the transcript. For fair comparison, we therefore identified human-annotated positions by selecting as many of the most likely positions in the distribution as our initial surprise-based event segmentation had identified in the transcript. In the same process used by Kumar et al. (2023), we then used their provided word onset times to translate these timestamps to token positions, allowing us to calculate our similarity metrics.

In Fig. 4B, we use Wasserstein distance in order to compare the relative positions of event boundaries between human annotations and those found by our own methods. Wasserstein distance is a versatile metric used to compare two probability distributions (Panaretos and Zemel, 2019). We used such a metric to better capture the uncertainty present in the human data, and found it to give more meaningful results than standard correlation or discrete distance metrics, which showed very little differences between methods. In order to calculate such a metric, we therefore need to convert our own discrete boundary positions to a distribution across token positions. We did so by defining a Mixture of Gaussians (MoG), with each Gaussian corresponding to a single position. Note that, for fair comparison with human data, we apply the same process to the discrete version of the human-annotated positions described above, and use this for comparison.

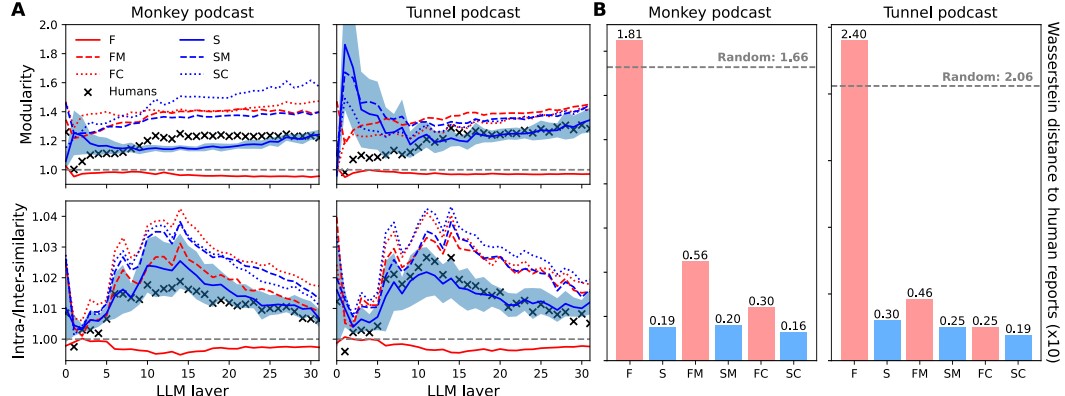

Figure 6: Comparison of human event segmentation with different computational segmentation methods in two human-annotated audio datasets. **(A)** Difference in metrics for the cohesion and separation of KV cache of LLaMA2 attention heads. The graphs report the difference of each method with the corresponding random segmentation. **(B)** Distance between human reports and different methods. In both sets of results, fixed methods (*F*, *FM*, *FC*) perform worse than their surprise-based counterparts (*S*, *SM*, *SC*) with InfLLM's method (*F*) performing worse than random.

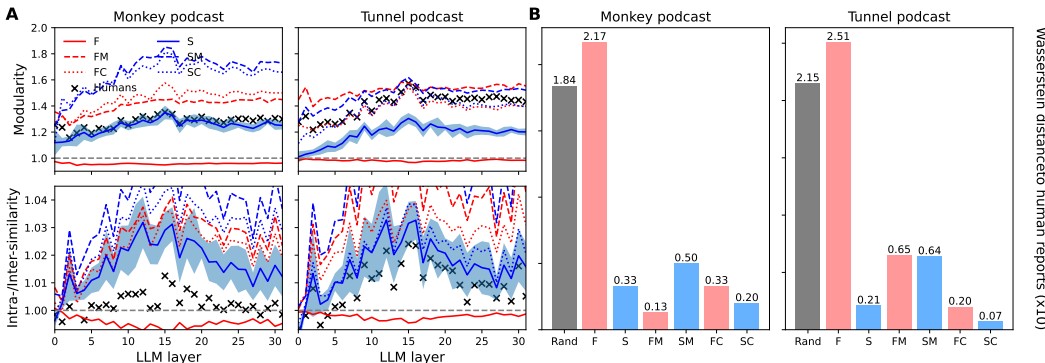

Figure 7: Comparison of human event segmentation with different computational segmentation methods using Mistral-7B. Plots include abbreviations like before.

## B.2 FURTHER RESULTS

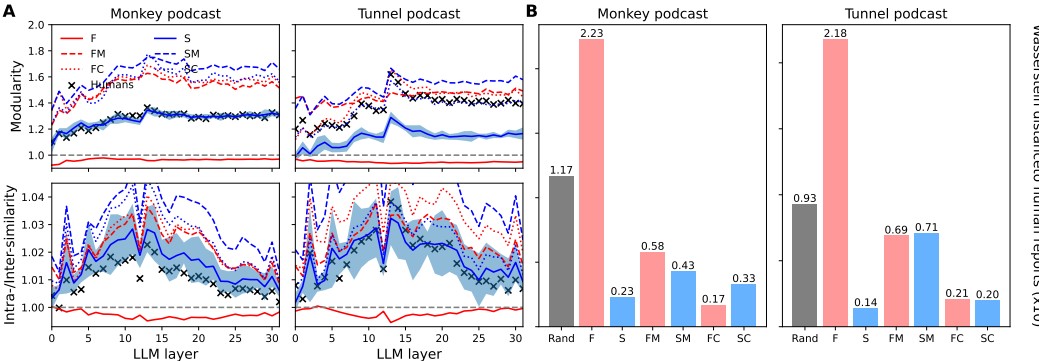

Figure 8: Comparison of human event segmentation with different computational segmentation methods using LLaMA-3-8B-Instruct. Plots include abbreviations like before.

## C  COMPLEXITY

### C.1  BOUNDARY REFINEMENT COMPLEXITY ANALYSIS

Here, we provide a detailed analysis of the computational complexity of our Algorithm 1, focusing on the boundary refinement step and the calculation of modularity and conductance metrics. Here we describe scaling complexity with example context length $n$, a chunk size $m$, and $k$ events.

**Metric Function Computation** The metric functions (modularity or conductance) are computed at the level of individual memory units but all rely on the same adjacency matrix for that chunk. The complexity of calculating the adjacency matrix, as defined in Eq. 2, is $\mathcal{O}(m^2)$. As the complexity for both metrics is largely dominated by, or on the same order as the computation of this term, the resulting complexity for the calculation of the metric function is $\mathcal{O}(m^2)$.

**Boundary Refinement Step** The boundary refinement step involves finding the optimal position $\hat{\beta}$ between each pair of consecutive initial boundaries $(\alpha, \beta)$ that optimizes the chosen metric function $f$. The iteration over initial boundaries has complexity $\mathcal{O}(k)$. For each updated boundary we change the community of one node at at time to the one on its right and re-evaluate the metric function $f$ with this change. Hence, for each boundary position, on average $2\frac{n}{k}$ nodes see a change in their community, and hence need to re-evaluate their contribution to the metric function. This results in a complexity of $\mathcal{O}(\frac{n}{k})$, for each position between $\alpha$ and $\beta$. This step will therefore scale with average event size resulting in a complexity $\mathcal{O}((\frac{n}{k})^2)$. Therefore, the overall complexity of this step scales with $\mathcal{O}(k(\frac{n}{k})^2) = \mathcal{O}(\frac{n^2}{k})$.

**Overall Complexity** The context is divided into $\frac{n}{m}$ chunks, and hence, we must compute this number of initial adjacency matrices with resulting complexity $\mathcal{O}(\frac{n}{m}m^2) = \mathcal{O}(nm)$. Adding in the complexity of the refinement updates, we get $\mathcal{O}(nm + \frac{n^2}{k})$. In practice, our method detects $k$ events such that $\frac{n}{k} \leq m$ is always true, hence this is upper-bounded by $\mathcal{O}(nm)$.

### C.2  ATTENTION COMPLEXITY ANALYSIS

**Attention.** Calculating a full attention matrix for a sequence length $n$ has complexity $\mathcal{O}(n^2)$. However, in the case of EM-LLM, we process such a context in smaller chunks of size $m$ and rely on retrieval to approximate a full attention matrix (also see Appendix F.1). Therefore we evaluate $\frac{n}{m}$ attention matrices each with complexity $\mathcal{O}(m(n_l + n_r))$ with $n_l$ the number of local tokens used and $n_r$ the number of tokens retrieved as part of the multi-stage attention process.

**k-NN Retrieval.** As previously mentioned, our approach relies on k-NN retrieval to avoid computing a full attention matrix. Such a retrieval process scales with the maximum number of memory units saved. In our case, as we enforce a minimum block size $b$, the maximum possible number of memory units to check for retrieval is $\frac{n-n_l}{b}$. As we retrieve memory units for every processed chunk (past $n_l$ processed tokens) this is upper-bounded by $\mathcal{O}(\frac{n(n-n_l)}{mb})$.

**Overall.** Including the retrieval step, overall our approach performs its attention computation with complexity $\mathcal{O}(n(n_l + n_r) + \frac{n(n-n_l)}{mb})$. In practice, for long-context tasks, this scales much slower than a full attention matrix, as visualized in Figures 9 and 10 which demonstrate this with our own settings and for values of $n$ up to 100K and 10M respectively. In fact, our complexity is negligible compared to full attention once sequence length reaches the millions.

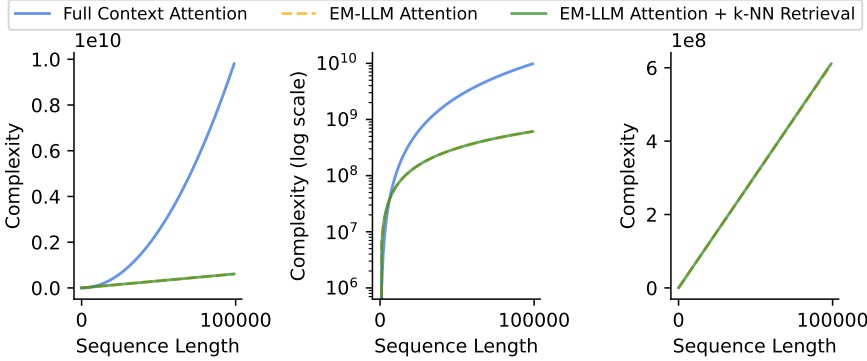

Figure 9: A visualization of the scaling complexity of EM-LLM vs a standard full-context approach (full attention matrix) as a function of sequence length (up to 100K tokens).

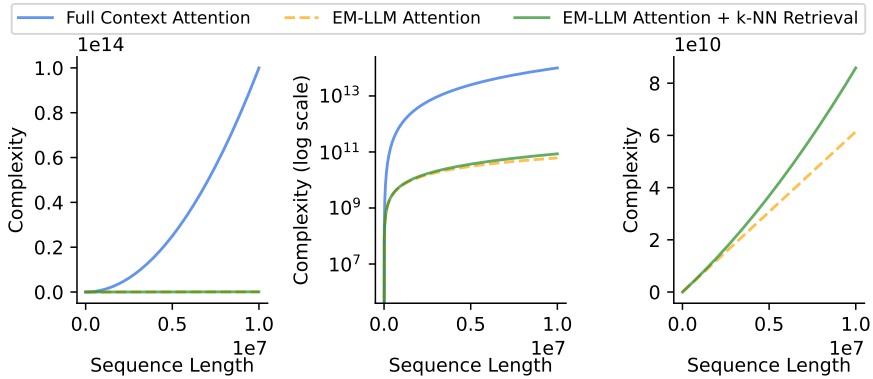

Figure 10: A visualization of the scaling complexity of EM-LLM vs a standard full-context approach (full attention matrix) as a function of sequence length (up to 10M tokens).

## C.3 IMPLEMENTATION AND USE OF HARDWARE RESOURCES

Retrieval-based methods for handling long sequence lengths can be especially appealing to those who may not have access to the hardware resources necessary to run large, long-context models. In this spirit, we describe here our adjustments to our framework made to further lower the minimum hardware requirements to accurately run inference on sequence lengths of 1M+ tokens. We note that our approach is very scalable in the sense that handling longer contexts only requires an increase in CPU memory, or even just disk storage.

All of our experiments were run on single nodes of 4 GPUs, each with 32GB of dedicated memory (except for the full-context results for which we used an API). Additionally, each node had a minimum of 100GB of CPU memory. While all base models and methods used across our own experiments fit on a single GPU, such hardware is still quite limited when compared to the more advanced H100s commonly used in research nowadays. Furthermore, the memory overhead due to processing and storing the KV cache and representative tokens for each memory unit for very long sequence lengths means that we have had to make some specific adjustments in order to ensure that we can efficiently run such experiments on older and limited hardware.

### C.3.1 WALL CLOCK TIME

As can be observed in Table 10, the largest increase to wall clock time in our framework is due to the similarity adjustment step. We would also like to note that such times increase steadily as sequence length increases, due to the increasing number of representative tokens involved in the

| | InfLLM | EM-LLM | | | |
| --- | --- | --- | --- | --- | --- |
| | | S | S+C | SM | SM+C |
| **Time** | 1.40s | 1.57s | 1.57s | 2.27s | 2.27s |
| **Ratio** | 1.0 | 1.12 | 1.12 | 1.62 | 1.62 |

Table 10: Difference in wall-clock time to process a single chunk of 512 tokens for various ablations of our framework as compared to InfLLM. Measured using Mistral-7B (4K+2K) and averaged over the first 100 chunks (51.2K tokens) of a long sequence.

k-NN calculation used for block retrieval, regardless of which method is used, although this is only noticeable when sequence length reaches the millions (see Appendix C.2).

### C.3.2  Memory Requirements

| Seq. Length | KV Cache | Rep. Tokens (max) |
| --- | --- | --- |
| 20K | 9.8GB | 0.6GB |
| 1M | 488.3GB | 30.5GB |

Table 11: Memory requirements for components of our framework which scale with sequence length, for a model with 32 layers and 32 KV heads, assuming half float precision and a batch size of 1. The number of representative tokens saved depends on the number of blocks, hence we show the maximum amount of memory required in the event the model segments the sequence into the maximum possible number of blocks.

As shown in Table 11, the KV cache can take up a lot of memory for longer sequences. While the bulk of it is kept on CPU memory until it is needed, such a resource is quickly spent when running multiple instances in parallel on a single node. Likewise, while the representative tokens for each block take up only a fraction of the memory compared to the KV cache, it can quickly become too large to keep on GPU memory. In order to address both of these issues and facilitate the use of our framework on lower-spec hardware, we have introduced various forms of offloading and model parallelisation.

CPU and Disk Offloading

InfLLM already provides details of their least recently used (LRU) KV cache management strategy used to efficiently offload the least-used memory units from GPU to CPU memory. We take inspiration from such a strategy and extend it to use allocated memory slots in CPU memory and dynamically offload such LRU units to disk space when the number of available slots runs low. Allocating and overwriting the memory slots, as is also done in the GPU cache, prevents our process from hanging on the memory which has been freed, avoiding system fragmentation and allocation errors. Furthermore, the ability to offload to disk means that our framework only requires approximately 2GB of CPU memory per instance in order to run with little overhead, as long as there is enough available disk space on the local machine.

Furthermore, we implemented the option to offload representative tokens to CPU memory for very long sequences (1M+), although we note that this further increases the overhead seen in wall clock time during k-NN retrieval.

Model Parallelisation

For very long sequences (2.5M+) it may be more time-efficient to parallelise the model's layers across multiple GPUs in order to keep representative tokens on the same GPU and speed up k-NN retrieval. We used Hugging Face's Accelerate to do this, along with some explicit casting of shared modules. Given a single node with a set number of GPUs and sequence length below 1M tokens, it is faster to offload the representative tokens and run a separate instance on each GPU, than to parallelise on 2 GPUs.

# D   FURTHER ABLATIONS

## D.1   HYPER-PARAMETER SELECTION AND TUNING

**Surprise Threshold Parameter** $\gamma$

Equation 1 introduces the surprise threshold parameter $\gamma$, which is responsible for the sensitivity of the threshold by scaling the standard deviation measured across the moving window. As such, with $\gamma = 1$, for a new token do be considered "surprising" it must achieve a surprise value greater than one standard deviation over the moving average. As mentioned in section 3.2, this ensures that the threshold adapts to contextual shifts therefore minimizing the need for manual tuning.

We initially explored our approach's sensitivity to $\gamma$ using Mistral on the LongBench benchmark. We evaluated the benchmark using surprise-only segmentation with $\gamma \in 1.0, 1.5, 2.0, 2.5, 3.0, 3.5$, as visualized in Figure 12. Naturally, we noticed that an increase in $\gamma$ resulted in a decrease in the number of events detected, and a resulting increase in the mean event size. In terms of overall performance, results suggest that a smaller $\gamma$, is generally best. We then explored such behavior across our segmentation methods, for which the overall behaviors were largely consistent with the surprise-only method. One particularly interesting observation is the fact the addition of our refinement algorithm (sM) seemed to show particularly high improvements over surprise-only at larger values of $\gamma$, although the best-performing value was still $\gamma = 1$.

Following these initial observations, in order to choose $\gamma$ for our experiments, we evaluated each model on the LongBench benchmark with $\gamma \in 1, 2, 3$ and surprise-only segmentation. We then selected the best-performing value of $\gamma$ for the rest of our experiments with each model. These were $\gamma = 1, 2, 1, 1, 1$ for Mistral, LLaMa-3, LLaMa-3.1, Phi-3, Phi-3.5 respectively, showing a consistent preference for $\gamma = 1$.

**Retrieved vs. Contiguity Buffer Ratio**

Section 3.4 introduces a similarity buffer of $k_s$ events and a contiguity buffer of $k_c$ events which retrieves $n$ events either side of each event in the similarity buffer. In total, the number of retrieved events is $k = k_s + k_c$. Note that we chose $n = 1$ for all experiments in order to balance contiguity with similarity. Moreover, Fig. 3A&B suggest that the contiguity effect is most significant for a handful of positions either side of the one recalled. This suggests that larger values of $n$ are likely to bring diminishing returns in terms of contiguity, while also reducing the size of the similarity buffer.

In practice, $k$ is expressed as a maximum number of tokens to ensure consistency as event sizes vary due to our dynamic segmentation methods. We therefore express the sizes of the similarity and contiguity buffers as a ratio $k_r = \frac{k_c}{k}$, and dynamically fill them with events such that $k_s = (1 - k_r)k$ and $k_c = k_r k$. In order to find the best contiguity ratio, we evaluated LongBench with Mistral and $\gamma \in 1, 2$ over values $k_r \in 0.3, 0.5, 0.7$, as illustrated in Fig. 13. To give a bit of intuition as to the meaning of these values, $k_r = 0.3$ will, on average, correspond to only including contiguous events for the top $25\%$ of events in the similarity buffer. On the other hand, $k_r = 0.7$, on average, will correspond to including contiguous events for all events in the similarity buffer, but will include $50\%$ less events in the latter. As mentioned in section 4.4, contiguity appears to have varying importance across tasks, with some performing best with the larger $k_r = 0.7$, but most did best with the lower $k_r = 0.3, 0.5$. Results also showed a clear preference for $\gamma = 1$, which is consistent with our previous experiments. Overall, there was a slight preference for $k_r = 0.3$, which we therefore selected for the rest of our experiments.

## D.2 SURPRISE, REFINEMENT AND CONTIGUITY

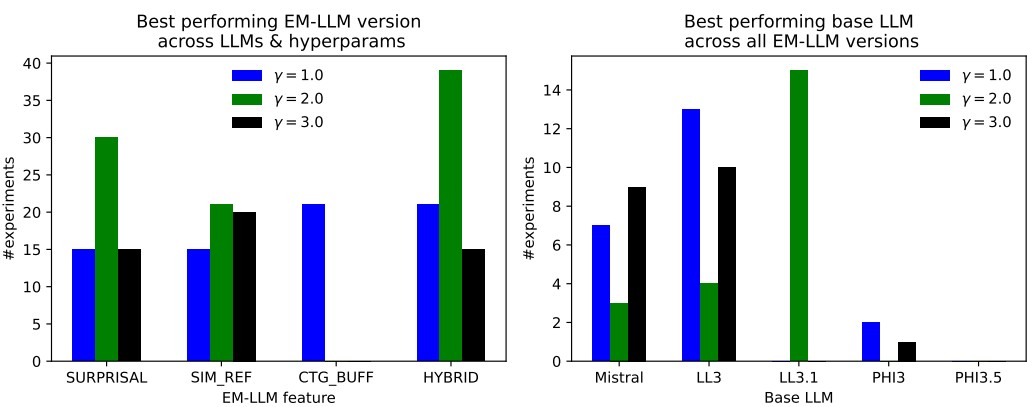

Figure 11: (left) Best performing version of the EM-LLM model across base language models (LMs) and hyper-parameters, including the parameter $\gamma$ that controls the sensitivity of the segmentation threshold in Equation (1). (right) Best performing base LM across all experiments presented in this work.

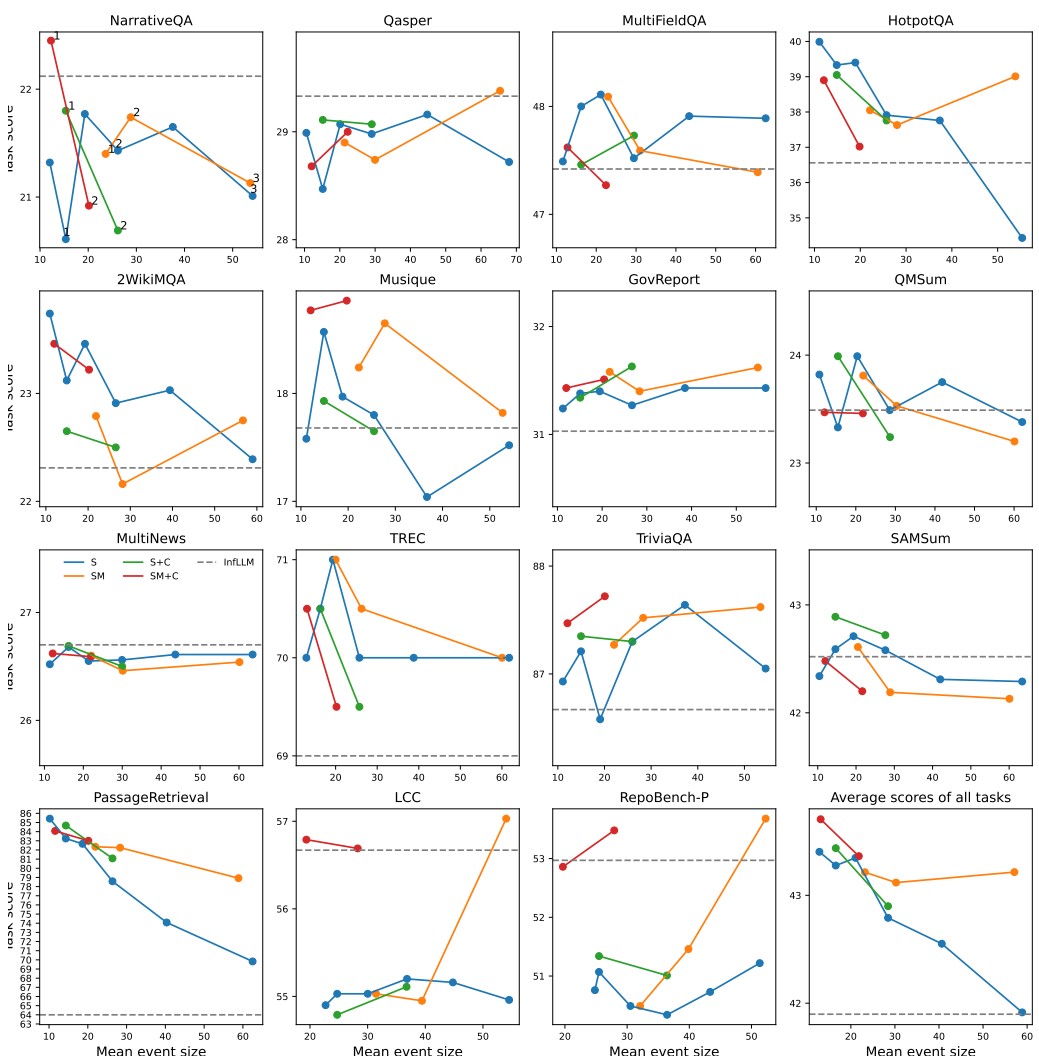

Figure 12: Ablation study in LongBench with Mistral-7B-Instruct-v0.2. Comparison of EM-LLM performance for different combinations of model features (represented by different colours) and different values of $\gamma$ (the threshold's scaling factor). Model variants are aligned on the x-axis based on the average number of block size that emerges for each case. The $\gamma$ values for each model variant are shown in the first sub-plot. The corresponding InfLLM performance is also shown.

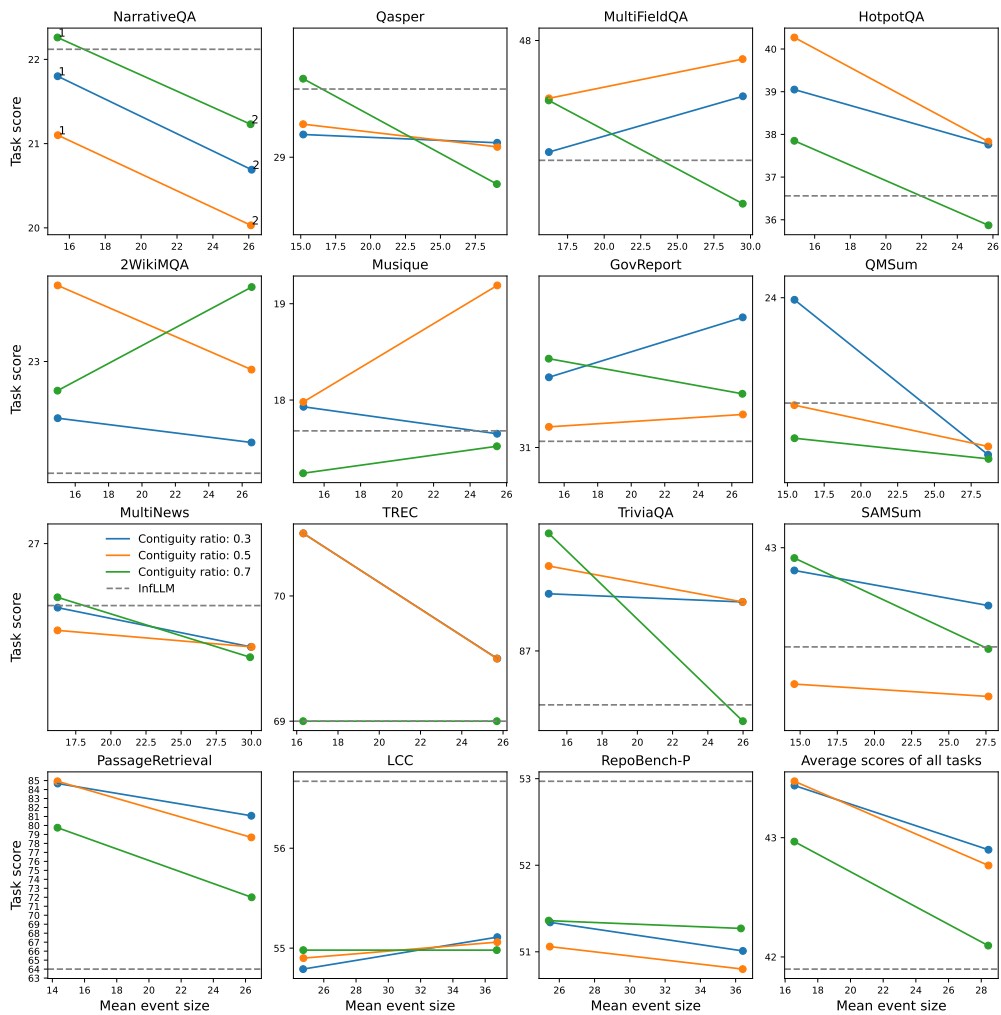

Figure 13: Ablation study in LongBench with Mistral-7B-Instruct-v0.2. Comparison of EM-LLM performance for different ratios of the contiguity and similarity buffers (represented by different colours) and different values of $\gamma$. Model variants are aligned on the x-axis based on the average number of block size that emerges for each case. The $\gamma$ values for each model variant are shown in the first sub-plot. The corresponding InfLLM performance is also shown.

## D.3 RETRIEVED TOKENS AND CONTEXT LENGTH

In our current experiments, we have chosen our buffer sizes to align with related works (namely InfLLM) in order to make direct performance comparisons. Such values also keep buffer sizes shorter than the average number of tokens in the evaluated benchmarks to ensure an appropriate use of retrieval. In order to further explore variations in these parameters, we ran a small ablation study varying the size of the retrieved buffer for summarization tasks in LongBench (Table 12 and Fig. 14). We have chosen such tasks as we believe they will be most likely to require more of the text content to give an accurate answer, and hence be most sensitive to the number of retrieved tokens.

However, for the following reasons, we believe this provides limited information on such parameters. For such a study, we would choose a base LLM trained with a relatively large context window, such as Mistral or LLaMa 3.1 which support context lengths of up to 32K and 128K respectively, in order to ensure that the underlying model can support an adequate range of buffer sizes. As LongBench may be considered relatively short compared to these context windows (average number

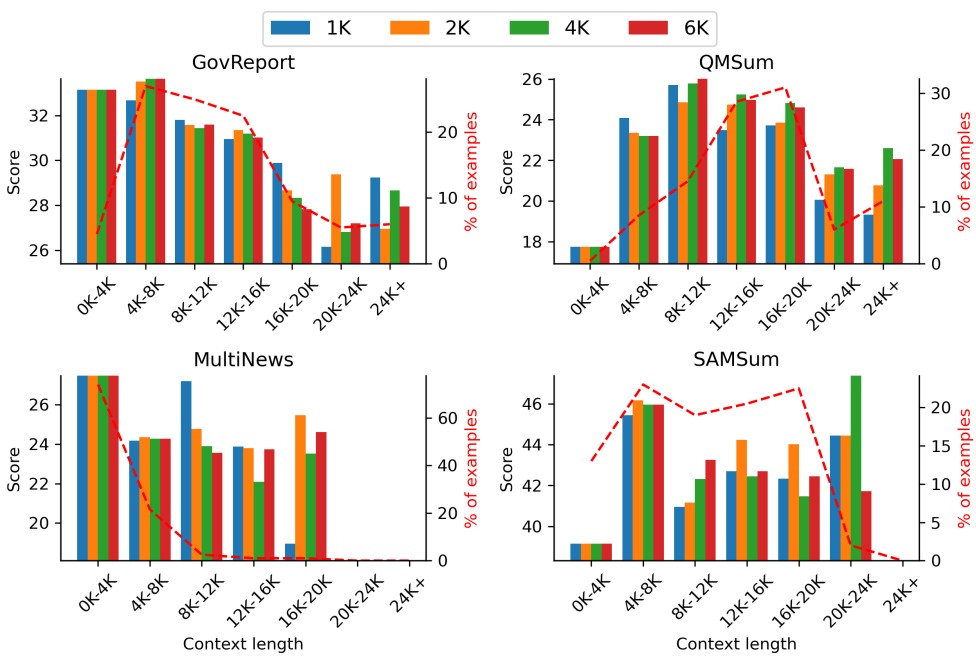

Figure 14: Ablation of the size of the retrieved buffer for EM-LLM$_S$ on Mistral with 4K local tokens on LongBench's summarization tasks. Presented as a function of context length.

of tokens per example: $12K \pm 10K$ with Mistral's tokenizer), $\infty$-Bench would be more appropriate (average number of tokens per example: $> 100K$). Unfortunately, evaluating larger buffer sizes (hence larger attention matrices) on the already-expensive $\infty$-Bench benchmark would be a very demanding ablation given our limited hardware resources, and hence we have left it for future work.

In the meantime, we provide the results mentioned below. Table 12 shows task-level performance is mostly consistent across ablations, although the "QMSum" task does seem to show some sensitivity to the number of retrieved tokens. This is further confirmed in Figure 14, which shows that longer examples benefit from more retrieved tokens. However, this is not the case in the other tasks, which seem to instead prefer less retrieved tokens. Furthermore, in these tasks, as examples get shorter the best performing number of retrieved tokens also seems to decrease. This is consistent with our observations concerning diluted attention and the decrease in accuracy when attending to too many tokens. Overall, such results, along with our positive results on the much longer $\infty$-Bench benchmark, further confirm that our approach is generally capable of efficiently handling context lengths much larger than the number of tokens available to the model at any one time.

| Task | 1K | 2K | 4K | 6K |
|---|---|---|---|---|
| GovReport | 31.26 | **31.44** | 31.33 | 31.26 |
| QMSum | 23.24 | 23.68 | **24.47** | 24.30 |
| MultiNews | 26.63 | **26.67** | 26.59 | 26.61 |
| SAMSum | 42.48 | **43.38** | 42.67 | 43.01 |

Table 12: Ablation of the size of the retrieved buffer (number of tokens) for EM-LLM$_S$ on Mistral with 4K local tokens on LongBench's summarization tasks.

# E    FURTHER DISCUSSION POINTS

## E.1    DETAILED ANALYSIS OF EM-LLM'S CONNECTION TO HUMAN EPISODIC MEMORY

This section provides a detailed analysis of how EM-LLM relates to human episodic memory (EM), addressing both the fundamental similarities and current differences between our computational approach and biological episodic memory systems.

### 1. Foundation: Transformers and Episodic Memory Integration

Recent work has shown that Transformer architectures naturally exhibit capabilities that parallel aspects of human episodic memory. In their basic operation, Transformers combine multiple pieces of information into coherent representations through latent embeddings - recollections of concepts that have been inferred from inputs and encoded in the embedding space. These concepts can be recalled and utilized via the SoftMax self-attention mechanism, enabling human-like behavior in short-context recall tasks (Ji-An et al., 2024).

However, two fundamental constraints prevent Transformers from maintaining this connection to human episodic memory in long-context scenarios: (a) The quadratic increase in computational and memory complexity and (b) the degradation of retrieval performance due to attention dilution Tworkowski et al. (2023); Ye et al. (2024).

### 2. EM-LLM's Approach to Human-like Memory Processing

Our architecture extends Transformers' inherent memory capabilities beyond these limitations through two key mechanisms that mirror human cognitive processes:

**2.A. Event-based Memory Formation**    We employ Bayesian surprise for event segmentation, a choice deeply grounded in both behavioural and neuroscientific evidence. Studies have shown that surprise signals in the hippocampus and other brain regions are crucial for event boundary and episodic memory formation (Sinclair et al., 2021; Zacks et al., 2007; 2011; Sherman et al., 2022; Mariola et al., 2022; Fountas et al., 2022). Our implementation demonstrates:

1. *Content-Dependent Parsing*: Analysis shows that tokens within surprise-based segments exhibit significantly higher key similarity than tokens across segments (Table 2), indicating natural semantic grouping that aligns with human event perception.

2. *Integration of Episode Components*: First, our model preserves temporal and contextual information within events, such as *when*, *where*, *what*, *how*, and *who*, as evidenced by strong performance on QnA and retrieval tasks. In addition, while our current implementation focuses on text, the architecture is fundamentally compatible with multi-modal processing. Like recent multi-modal models, including Qwen2-VL (Wang et al., 2024), Pixtral (Agrawal et al., 2024), LLaMa 3.2, EM-LLM can integrate different modality encoders into a single embedding space, treating all embeddings equally in the KV cache.

**2.B. Human-like Information Retrieval**    Our two-stage retrieval process combines both similarity-based retrieval for cued recall and decaying temporal contiguity reflecting free recall patterns. This integration enables our model to exhibit both temporal contiguity effects and temporal asymmetry in recall - behavioural patterns consistently observed in human EM retrieval studies (Howard and Kahana, 2002). The inclusion of the contiguity buffer specifically allows for the maintenance of temporal relationships in a way that mirrors human memory access patterns. Moreover, retrieval is done individually per-layer further supporting the transformer's learned ability to focus on different aspects of the sequence (see Appendix D.3), including necessary contextual information. The combination of contiguity and layer-wise retrieval results in a sophisticated information retrieval process which, combined with a transformer, has all the tools to allow for the complete contextual recollection of relevant events.

### 3. Current Limitations and Future Directions

While EM-LLM successfully implements key aspects of human episodic memory, several important differences remain:

1. *Non-parametric nature*: Unlike human memory, which involves complex synaptic weight changes, our method relies on non-parametric storage of key-value pairs.

2. *Hierarchical event structure*: Current implementation lacks the sophisticated nested event representations observed in human cognition (Baldassano et al., 2017).

3. *Cross-Modal integration*: While architecturally compatible, our current implementation doesn't fully capture the rich multi-modal integration characteristic of human episodic memories.

4. *Memory Consolidation*: The model lacks mechanisms for long-term memory formation processes and systems consolidation observed in biological memory systems.

These limitations represent opportunities for future work rather than fundamental flaws in our approach. A parametric version of EM-LLM could help reduce LLMs' memory complexity to a bare minimum by replacing the vector database and KV cache storage requirements with a neural approach (e.g., using the model in Ramsauer et al., 2021). Additionally, developing hierarchical event representations and integrating memory consolidation mechanisms could facilitate continual learning and further bridge the gap between artificial and biological episodic memory systems. In conclusion, we believe to have provided strong arguments that our approach is capable of complete recollections of experiences, resembling EM in humans. However, we acknowledge that we lack explicit empirical evidence that this is how the resulting model makes use of the architecture in order to achieve the results presented, and hence clarify that we only claim an EM-inspired approach, rather than an actual human-like EM process.

## 4. Relationship to Different Memory Systems

While our work draws primary inspiration from episodic memory systems, the relationship between different types of memory (episodic, semantic, working memory, and other systems) is complex and often overlapping, both in biological and artificial systems. Our approach focuses specifically on episodic memory-like features such as event segmentation and temporal organization of experiences. However, as discussed in Section 5, our architecture also shows interesting parallels to working memory models, particularly in how the local context aligns with concepts like Baddeley's working memory system and Cowan's focus of attention. While some aspects of our model, particularly the learned representations in the underlying LLM, may share characteristics with semantic memory systems, the primary innovations in EM-LLM centre on episodic-like features. Future work could explore these connections more explicitly, potentially leading to architectures that better capture the interactions between different memory systems, including the development of modality-specific buffers inspired by Baddeley's multi-component model.

### E.2 Architecture contributions of EM-LLM

EM-LLM introduces three novel architectural contributions for LLMs, for which we have shown their importance both conceptually and with empirical results.

(1) **Dynamic surprise-based segmentation**. The method to segment a given context window based on Equation (1) is the first method for dynamic segmentation of KV cache into blocks, and also the first that manipulates the KV cache based on insights from cognitive science, using an intrinsic measure to LLMs. We show empirically using multiple LLMs that this low-cost and simple-to-implement method is able to group relevant pairs of keys and values (KV) together (relevance measured as key similarity) with much higher accuracy than fixed segmentation, the only alternative proposed approach (See Table 2 for key similarity comparisons). We also show that this method results in increased LLM performance, especially in retrieval tasks (16.6% average increase over InfLLM) and multi-document QA tasks (6.4% average increase over InfLLM) across all the LLMs we tried (See the "S" column in the tables of Appendix A.1).

(2) **Graph-based refinement**. The method presented in Algorithm 1 is the first to refine the temporal borders of events in the context window of LLMs using graph theory. We relied on the insight that tokens are more useful to be recalled together, if the variance between their keys is low, as they need to be used by a single query at the time. This method can also stand by itself as a dynamic segmentation approach of KV cache, more computationally heavy than surprise-based segmentation but achieving a competitive accuracy in grouping relevant (KV) together (see again Table 2), while it has the extra benefit that can be used in each attention head independently, without relying on the LLM output.

(3) **Contiguity buffer**. This is a dedicated decaying buffer in the context window of LLMs that maintains the KV cache of temporally contiguous tokens to the context window of the LLM for a certain amount of time. This relies on the recent insight that self-attention heads responsible for in-context learning are shown to consecutively attend to contiguous groups, similarly to human studies (Ji-An et al., 2024). We show that this algorithm can also be combined with methods (1)

and (2) and results in further increases in the overall LLM performance. Notably, the average increase in retrieval tasks over InfLLM jumps to $19.4\%$, and for multi-document QA tasks to $9.2\%$ across all the LLMs we tried (See the "SM+C" column in the tables of Appendix A.1).

### E.3 Why is the Refinement Algorithm Effective?

The use of the argmax in Algorithm 1 guarantees either a positive improvement or no change in similarity for each event boundary position update, hence either improving overall similarity or showing no change from surprise-based segmentation. Therefore, while we would ideally find the globally optimal event boundaries with regards to the similarity metric, and seek to converge to this point, this would be much more expensive to compute and introduce a lot of overhead for every processed chunk of the context and the corresponding memory units. Instead, our algorithm simply implements a cost-effective way to look for *any* potential increase to this metric, as it has been empirically shown to do successfully in section 4.2. Nevertheless, to briefly touch on the convergence of such a method, our approach can be seen as a single pass of Phase 1 of the heuristic Louvain method (Blondel et al., 2008) initialized with surprise-based segmentation (as opposed to having each node assigned its own community), and modified to only consider the move of a node to its right-side neighboring community. As our initial results had shown that surprise-based segmentation already achieves higher similarity metrics (including modularity, which is the objective used in the Louvain method) than fixed or random segmentation (Table 2), we believe this is a good initialization as it means that our algorithm will, at worst, achieve the same similarity metrics. While the Louvain method is considered to be an efficient way to converge to *local* optima when iterated, our own modifications and lack of iterations mean we cannot claim such behavior but rather suggest that we are likely to see some improvements in our metrics, as our results have confirmed.

### E.4 Feasibility of End-to-End Neural Implementations

A significant difference between EM-LLM and biological memory systems is the ability of neural circuits in the brain to learn and adapt their event segmentation and memory retrieval mechanisms through experience. In artificial neural networks, this would correspond to end-to-end optimization via differentiable architectural components. Below, we discuss the feasibility of such an approach and compare it with our current implementation:

**Event segmentation:** Differentiable event segmentation models have already demonstrated the feasibility of learning a temporal structure from continuous experience. Models like SEM (Franklin et al., 2020) show how neural networks can combine with probabilistic inference to capture human-like event segmentation, while approaches like the DLH (Zakharov et al., 2022b) demonstrate that neural architectures can learn to identify hierarchical temporal boundaries through differentiable clustering and amortised variational inference. For instance, using the VaDE trick in (Jiang et al., 2016). These approaches offer powerful advantages in terms of learned representations and flexibility, potentially capturing the complex hierarchical event structure of real environments and adapting to different domains. Particularly compelling advantages include the ability to perform layer-wise or attention-head-wise segmentation and the potential emergence of nested timescale structures, as demonstrated in (Zakharov et al., 2022a;b), mirroring how the brain processes events at multiple temporal scales (Baldassano et al., 2017). While such end-to-end training is theoretically appealing and mirrors how neural circuits might learn temporal structure, our method takes a more pragmatic approach by leveraging the pre-trained capabilities of LLMs. By using Bayesian surprise computed directly from model outputs to detect event boundaries, we achieve efficient segmentation without requiring complex architectural modifications or additional training, while still aligning with cognitive theories about prediction errors in event perception (Zacks et al., 2007).

**Retrieval:** The development of neural architectures for memory retrieval has evolved from classical Hopfield networks (Hopfield, 1982) through several key innovations. Early Hopfield networks demonstrated how content-addressable memory could emerge from simple neural circuits, paralleling biological memory systems. This was significantly advanced by Neural Turing Machines (Graves, 2014) and their successor, the Differentiable Neural Computer (Graves et al., 2016), which introduced differentiable memory access mechanisms. Modern Hopfield networks (Ramsauer et al., 2020) further revolutionized our understanding by establishing a theoretical connection between transformer attention and associative memory, showing how these systems can store and retrieve exponentially many patterns while maintaining stable dynamics. Such end-to-end approaches could particularly benefit the quality of memory representations, as they could learn optimal projections for generating

representative keys for memory blocks, potentially capturing universal contextual patterns more effectively than our current approach. While end-to-end training of memory systems is feasible, as demonstrated by models like MERLIN (Wayne et al., 2018), such approaches often face challenges with credit assignment over long sequences and require complex architectural modifications. Our KNN-based approach leveraging the KV cache offers a pragmatic middle ground: it harnesses the rich semantic representations already present in transformer models while maintaining the computational benefits of nearest-neighbour retrieval. This aligns with both biological intuitions about pattern matching in the hippocampus (O'Reilly and Norman, 2002) and the theoretical foundations of modern Hopfield networks, where similarity-based attention serves as a form of associative memory. By operating on pre-trained representations, our method sidesteps the training complexities of fully differentiable memory while preserving the benefits of content-based retrieval.

**Refinement:** The refinement of event boundaries could also theoretically be learned end-to-end, similar to how attention pruning mechanisms (Ying et al., 2019) learn to identify optimal subgraphs in graph neural networks, or how hierarchical clustering can be made differentiable (Ying et al., 2018). Our graph modularity approach provides a computationally efficient alternative that optimizes for coherence within segments while respecting the initial surprise-based boundaries. While our method is primarily motivated by computational considerations, it parallels how memory consolidation might strengthen associations between related elements within an event while weakening cross-event associations (Preston and Eichenbaum, 2013). The high modularity of our surprise-based segmentation, even before refinement, suggests that prediction errors naturally tend to occur at boundaries between coherent event structures.

### E.5 FUTURE EXTENSIONS INSPIRED BY HUMAN MEMORY SYSTEMS

Human episodic memory exhibits several sophisticated features beyond those currently implemented in EM-LLM. Here, we discuss how incorporating these additional characteristics could enhance our model's capabilities and performance:

**Hierarchical organisation:** A hierarchical structure in memory can provide multiple advantages such as improved retrieval, more disentangled latent embeddings, longer future predictions (Saxena et al., 2021; Zakharov et al., 2022b), better planning (Hafner et al., 2022) and higher agreement with neural processes in the brain Baldassano et al. (2017). In our model, a hierarchical organisation of episodic memories based on the existing hierarchy of embeddings in the LLM layers could be implemented by extending our segmentation processes to operate at each layer of the Transformer independently. This could be achieved either through a differentiable approach or a layer-specific surprise metric. Interestingly, our current k-NN retrieval approach already implicitly leverages hierarchical structure through its underlying approximate nearest neighbour algorithms, which typically employ tree-based structures (Johnson et al., 2019) to efficiently partition the embedding space.

**Memory consolidation:** The brain's process for memory consolidation is crucial for continual learning, an ability that remains largely unsolved in current LLMs. Implementing consolidation mechanisms in EM-LLM could help address catastrophic forgetting while enabling more efficient integration of new information with existing knowledge.

**Mental time travel:** The ability to employ the same retrieval mechanism for imagining future events as for recalling past experiences is a key feature of episodic memory that could significantly enhance LLMs' planning and reasoning capabilities. By leveraging its event-based structure to simulate potential future scenarios or recall past experiences in novel contexts, this mechanism could provide a powerful solution for planning and reasoning, which are currently important challenges in large generative models.

## F PROOFS

### F.1 APPROXIMATE EQUIVALENCE OF K-NEAREST NEIGHBOURS AND SOFTMAX ATTENTION

Here we will attempt to show that using a k-NN retrieval in a key-value cache as part of the attention mechanism in transformers is an approximation of applying softmax attention over the entire sequence of tokens.

Let $q$ be a query vector and $K = \{k_1, k_2, \ldots, k_n\}$ the set of key vectors in a transformer model with dimensionality $d$. Each key $k_i$ has a corresponding value vector $v_i$, with $V = \{v_1, v_2, \ldots, v_n\}$. The softmax attention weights $a_i$ are defined as:

$$a_i = \frac{\exp(q \cdot k_i \, d^{-\frac{1}{2}})}{\sum_{j=1}^{n} \exp(q \cdot k_j \, d^{-\frac{1}{2}})} \tag{6}$$

The output vector $u$ is computed as:

$$u = \sum_{i=1}^{n} a_i v_i \tag{7}$$

In the k-NN approach, a subset $K'$ of size $k$ is selected, containing keys nearest to $q$. The approximated attention weights $a_i'$ over this subset are:

$$a_i' = \frac{\exp(q \cdot k_i \, d^{-\frac{1}{2}})}{\sum_{j \in K'} \exp(q \cdot k_j \, d^{-\frac{1}{2}})} \quad \text{for } k_i \in K' \tag{8}$$

The approximate output vector $u'$ is:

$$u' = \sum_{k_i \in K'} a_i' v_i \tag{9}$$

ASSUMPTIONS

1. *Exponential Dominance*: The exponential function in the softmax is sharply peaked, implying that keys with the highest similarities to $q$ contribute significantly more to the sum than others.

2. *Representativeness of k-NN Subset*: The subset $K'$ captures the majority of the attention weight from the full set $K$.

**Lemma 1: Dominance of k-NN Subset**   If $K'$ consists of the $k$ keys with the highest dot products $q \cdot k_i$, then:

$$\sum_{j \in K'} \exp(q \cdot k_j \, d^{-\frac{1}{2}}) \geq \alpha \sum_{j=1}^{n} \exp(q \cdot k_j \, d^{-\frac{1}{2}}) \tag{10}$$

for some $\alpha \approx 1$, typically very close to 1.

**Proof**: This follows from the exponential dominance assumption and the nature of the exponential function, which is sharply peaked.

**Lemma 2: Approximation of Output Vector**   Given the dominance of $K'$ as shown in Lemma 1, the approximate output $u'$ effectively represents the full output $u$:

$$\|u' - u\| \leq \epsilon \tag{11}$$

where $\epsilon$ is a small error term.

**Proof**: Follows from the weighted sum structure of $u$ and $u'$, using the bounds established in Lemma 1.

Given the lemmas and under the stated assumptions, the k-NN retrieval mechanism within a key-value cache effectively approximates the softmax attention mechanism in transformers. This proof highlights the efficiency versus accuracy trade-off inherent in using approximate methods like k-NN retrieval.

