# OpenReview forum: "Human-inspired Episodic Memory for Infinite Context LLMs"
_ICLR.cc/2025/Conference — ICLR 2025 Poster_

### Official Review · Reviewer_N12r · 2024-10-24

**Soundness:** 2
**Presentation:** 2
**Contribution:** 2
**Rating:** 5
**Confidence:** 2

**Summary:**

This paper introduces EM-LLM, a novel approach that integrates key aspects of human episodic memory and event cognition into LLMs with no fine-tuning, enabling them to handle practically infinite context lengths while maintaining computational efficiency

**Strengths:**

They introduced EM-LLM, a flexible architecture that integrates key aspects of human episodic memory and event cognition into Transformer-based LLMs. Their approach enables existing LLMs to effectively process vastly extended contexts without the need for pre-training. They perform successful passkey retrieval across 5M tokens, a length which is computationally infeasible for current full-context models

**Weaknesses:**

1. As you mentioned, almost all the methods used in your paper appear to have been proposed by others. Could you clarify what specific innovations you are introducing?

2. You stated that your method outperforms others, but in Table 1, I did not see the impressive results you described. It seems that your method may not be as effective as claimed.

3. Do you apply Eq. (1) to each token individually? If so, I imagine this would require substantial processing time. How do you address this issue?

4. In Eq. (2), could you explain how you derive the key vectors?

5. You employ k-NN in the MEMORY RETRIEVAL stage—does this significantly increase computation time?

6. I found Figure 4 difficult to understand. Could you provide additional explanation to clarify it?

**Questions:**

The same to Weaknesses.

---

> ### Author Response · Authors · 2024-11-19
> **(Answer 1/2)**
>
> We would like to thank reviewer N12r for their questions. We have updated our manuscript to address all points raised. Below, we provide detailed responses to each concern and we remain open to any additional feedback that could further improve our work.
>
> ### Questions:
>
> * **Reviewer:** _1. As you mentioned, almost all the methods used in your paper appear to have been proposed by others. Could you clarify what specific innovations you are introducing?_
>
>   **Authors:** We have made amendments to our paper to clarify this very important point, which we summarise here. In this work we have made *three* novel architectural contributions for LLMs, for which we show their importance both conceptually and with empirical results:
>
>      1. **Dynamic surprise-based segmentation.** We introduce the first method for dynamic segmentation of KV cache into blocks. Our method is also the first that manipulates the KV cache based on insights from cognitive science, using an intrinsic measure to LLMs. We show empirically using multiple LLMs that this low-cost and simple-to-implement method is able to group relevant pairs of keys and values (KV) together (relevance measured as key similarity) with much higher accuracy than fixed segmentation, the only alternative proposed approach (See Table 2 for key similarity comparisons). We also show that this method results in increased LLM performance, as mentioned in our next answer.
>
>      2. **Graph-based refinement.** We were the first to introduce an algorithm to refine the temporal borders of events in the context window of LLMs using graph theory. We relied on the insight that tokens are more useful to be recalled together, if the variance between their keys is low, as they need to be used by a single query at the time. This method can also stand by itself as a dynamic segmentation approach of KV cache, more computationally heavy than surprise-based segmentation but achieving a competitive accuracy in grouping relevant (KV) together (see again Table 2), while it has the extra benefit that can be used in each attention head independently, without relying on the LLM output.
>
>      3. **Contiguity buffer**. We were the first to introduce a method to maintain a dedicated decaying buffer in the context window of LLMs that maintains the KV cache of temporally contiguous tokens to the context window of the LLM for a certain amount of time. For this, we relied on the recent insight that self-attention heads responsible for in-context learning are shown to consecutively attend to contiguous groups, similarly to human studies (Ji-An et al., 2024). We show that this algorithm can also be combined with methods (1.) and (2.) and results in further increases in the overall LLM performance (see again next answer).
>
> * **Reviewer:** _2. You stated that your method outperforms others, but in Table 1, I did not see the impressive results you described. It seems that your method may not be as effective as claimed._
>
>   **Authors:** Thank you for the question. Significant performance benefits of our model are particularly evident for QnA and retrieval tasks, and they can also largely be found when only our dynamic surprise-based segmentation is applied. In particular, surprise-based segmentation results in $16.6$% average increase over InfLLM in retrieval tasks, and $6.4$% average increase in multi-document QA tasks across all the LLMs we tried. In addition, when graph-based refinement is also added, this average performance increase jumps to $19.4$% for retrieval tasks, and to $9.2$% for multi-document QA tasks (See the columns "S" and "SM+C" in the tables of Appendix A.1). To further support our answer, we also ran statistical tests to explore whether the improvements we see over InfLLM on a benchmark level are statistically significant. The vast majority of our results (all base LLMs except phi-3.5) show a statistically significant improvement over InfLLM at the benchmark level ($p < 0.05$ using a two-tailed z-test). This analysis is now appended to our manuscript.
>
>   Furthermore, we demonstrate broad performance benefits beyond InfLLM. Our model is the first to our knowledge to achieve superior performance compared to the state-of-the-art RAG retriever NV-Embed-v2 without requiring costly external architectural components such as larger LLMs ($36.4$% vs $51.6$% in all LongBench tasks; see Figure 1 and Table 9). The computational complexity of our method is strikingly lower than processing the complete context within the LLM, while maintaining competitive performance across tasks and surpassing full-context processing in the majority of long-context tasks ($39.3$% vs $51.6$% in LongBench; see again Figure 1). Most notably, we achieve perfect retrieval accuracy on 10M tokens while using only a single GPU - a first in the field. These characteristics establish both the novelty and practical relevance of our approach, especially since it can be applied to any LLMs without further training.

---

> ### Author Response · Authors · 2024-11-19
> **(Answer 2/2)**
>
> * **Reviewer:** _3. Do you apply Eq. (1) to each token individually? If so, I imagine this would require substantial processing time. How do you address this issue?_
>
>     **Authors:** While the calculation of surprise and corresponding boundaries does scale with O(n), where n is the number of input tokens, the probability in the left-hand side of the inequality is the output of the base LLM. Hence, this calculation boils down to a simple moving average which is largely negligible compared to a single forward pass of the base LLM. As a result, we have not noticed any impact on processing time.
>
> * **Reviewer:** _4. In Eq. (2), could you explain how you derive the key vectors?_
>
>     **Authors:** Unless we have misunderstood your question, we have not derived the key vectors ourselves but rather used standard notation for the key vectors as part of the Transformer architecture. In this case, these are simply the keys for the input sequence output by each head in question, as described in the paragraphs around the equation. However, please do let us know if there has been some confusion here, or if you would like more information and we will be happy to address it.
>
> * **Reviewer:** _5. You employ k-NN in the MEMORY RETRIEVAL stage—does this significantly increase computation time?_
>
>     **Authors:** As you have correctly pointed out, k-NNs does impact computation times during the retrieval stage. However, as briefly mentioned in Appendix C.2, which discusses complexity and wall clock time, this only becomes significant in very long contexts (millions of tokens). Furthermore, as mentioned in Section 3.4 of the main text, one may use approximate k-NNs in order to speed this up and maintain efficiency. We also feel that it is important to note that alternatives such as RAG also rely on k-NNs, while full-context methods are much more computationally demanding than our entire approach at context lengths where k-NN retrieval may just become noticeable for us, even without using approximate k-NNs. We will expand on these points in the sections of the paper mentioned and add figures to visualise such overhead as a function of context length for both our approach and full-context models. A more in-depth analysis of such overhead in terms of wall-clock time could be a somewhat interesting addition to the paper but given the overwhelming use of modern GPUs in LLMs, we feel this would be mostly reflective of our hardware, while more general time and computational complexity analysis for full and approximate k-NNs has been vastly covered already in the literature.
>
> * **Reviewer:** _6. I found Figure 4 difficult to understand. Could you provide additional explanation to clarify it?_
>
>     **Authors:** While more details explaining how the results for this specific figure were calculated are available in Appendix B.1, we fully agree that the results themselves should be clear without it and would be more than happy to clarify this figure. We have amended the caption of this figure based on Reviewer _pMUj_. Could the reviewer please verify if the figure is clarified now and, if not, be more specific as to which sub-plot they find confusing and why?
>
>
> ### Summary of changes made to the paper based on your feedback:
> * Question 1:
>     1. We have updated the last two paragraphs of the **Introduction** to clarify our contributions.
>     2. We have introduced the **Appendix E.2** section to further clarify this by reiterating the points in this response.
> * Question 2:
>     1. We added discussion on significance to Appendix A.1.
>     2. We also added the points from this answer to the **Appendix E.2** mentioned in Question 1.
> * Question 5:
>     1. We added a section to **Appendix** providing complexity analysis and figures to visualise scaling behaviour for k-NNs within our approach, as well as our overall approach compared to standard full-context models for the same sequence length.
> * Question 6:
>     1. We have updated the caption of **Figure 4**.

---

> > ### Comment · Reviewer_N12r · 2024-11-28
> >
> > I apologize for the delay in providing my review comments. Initially, I stated that I am not an expert in your field, and I found your manuscript quite challenging to read. If you feel that you have received an unfair review, I am truly sorry. However, I have also experienced situations where reviewers provided malicious negative reviews without any response. Therefore, if you believe there are issues with my review, please feel free to bring them to the attention of the AC. Besides, I truely think you should polish your paper.
> >
> > Thank you for your understanding.

---

### Official Review · Reviewer_RkzP · 2024-10-29

**Soundness:** 2
**Presentation:** 3
**Contribution:** 2
**Rating:** 5
**Confidence:** 4

**Summary:**

This paper introduces EM-LLM, a method for extending the context window of LLMs by incorporating principles from human episodic memory and event cognition. The method segments input sequences into episodic events using a combination of Bayesian surprise and graph-theoretic boundary refinement. It implements a two-stage memory retrieval process that combines similarity-based and temporally contiguous retrieval. The authors evaluate their method on LongBench and InfiniteBench benchmarks, comparing against state-of-the-art retrieval models like InfLLM and RAG.

**Strengths:**

The paper proposes the idea of equipping LLMs with cognitive science principles (episodic memory and event cognition), which aligns well with current challenges in long-context processing;
The use of Bayesian surprise for dynamic event segmentation in LLMs is novel, moving beyond simple fixed-length segmentation used in prior work like InfLLM;
The method achieves 100% accuracy on Passkey.Retrieval task with sequences up to 5M tokens, demonstrating practical scalability;
The paper provides a scalable alternative to both RAG and full-context approaches, outperforming them on most tasks while maintaining computational efficiency, increasing its practical value.

**Weaknesses:**

1.The paper claims surprise-based segmentation is superior to fixed-length approaches (like InfLLM), but lacks theoretical justification for why this should be effective. While Figure 4 shows some empirical correlation with human segmentation, there's no analysis of why this leads to better LLM performance. The authors should provide a theoretical analysis showing why this metric captures semantically meaningful boundaries better than alternatives.
2.The boundary refinement process (Algorithm 1) lacks convergence analysis. Given that it's iteratively adjusting boundaries based on modularity/conductance metrics, it is important to establish guarantees about stability and convergence.
3.The two-stage memory retrieval process (Section 3.4) introduces ks and kc parameters for similarity and contiguity buffers, but it does not justify the choice of values or analyze their trade-offs.
4.Although Table 1 shows some improvements over InfLLM, the improvements on most tasks may not be statistically significant.
5.The human correlation study in Section 4.2 relies on 3 short podcasts, which is insufficient to make broad claims about alignment with human event segmentation. The authors should expand the section to a larger, more diverse dataset of human annotations.
6.Section 4.4 discusses the impact of contiguity but does not properly ablate different buffer sizes or analyze the trade-off between similarity and contiguity buffers.

**Questions:**

1.How sensitive is the method to the choice of the surprise threshold parameter? What is the process for tuning this parameter?
2.How do you ensure the stability of the boundary refinement process? Are there cases where it fails to converge?
3.Does the issue of needing to query too many matching events still exist when the context window is too long?
4.If we use the Surprise-based method to segment the context into a RAG format, will the performance remain the same?
5.How does the method handle dynamic changes in context relevance over time?

---

> ### Author Response · Authors · 2024-11-19
> **(Answer 1/4)**
>
> We would like to thank reviewer RkzP for their careful review and thoughtful questions, which have helped us identify areas where we needed to clarify our contributions and strengthen our arguments. We have updated our manuscript to address all points raised. Below we provide detailed responses to each concern. We remain open to any additional feedback that could further improve our work.
>
> ### Responses to reviewer's questions:
>
> * **Reviewer:** _How sensitive is the method to the choice of the surprise threshold parameter? What is the process for tuning this parameter?_
>
>     **Authors:** While we have already included figures in Appendix D which illustrate ablations for the threshold scaling factor, relative buffer sizes, and method used, we agree that these decisions and their supporting observations should be included in the paper. In this particular case we have carried out a short ablation study on each model with suprise-only segmentations to select the best value to use for our experiments. Although there is a consistent preference for certain values across all models, results suggest that the method is largely insensitive to this parameter in the sense that minor variations in its value do not significantly affect downstream performance. We now mention this in the main text and provide more details in the Appendix.
>
>     * **Summary of changes made to the paper:**
>         1. Pointed to **Appendix D** in the caption for **Table 1**.
>         2. Added a sub-section to **Appendix D** providing further details for our initial ablations and parameter tuning.
>
> * **Reviewer:** _How do you ensure the stability of the boundary refinement process? Are there cases where it fails to converge?_
>
>     **Authors:** This is a very pertinent observation, thank you. Through our own omission, there appears to have been some confusion as to the implementation of the algorithm which is relevant to the need for convergence and stability analysis. Namely, there are no iterations for single boundaries, as opposed to what our statement on line 313 of the main text appeared to suggest. We simply loop through each event boundary and perform a **single** update using an argmax function to measure the best (new) position for that boundary within the event to its left, and with respect to the overall similarity metric of the current processed sequence. After this update, the corresponding memory units are saved and no longer modified. This guarantees either a positive improvement or no change in similarity for each event boundary position update, hence either improving overall similarity or showing no change from surprise-based segmentation. Therefore, while we would ideally find the globally optimal event boundaries with regards to the similarity metric, and seek to converge to this point, this would be much more expensive to compute and introduce a lot of overhead for every processed chunk of the context and the corresponding saved memory units. Instead, our algorithm simply implements a cost-effective way to look for **any** potential increase to this metric, as it has been empirically shown to do successfully in section 4.2. Nevertheless, to briefly touch on the convergence of such a method, our approach can be seen as a single pass of Phase 1 of the heuristic Louvain method (Blondel et al, 2008) initialised with surprise-based segmentations (as opposed to having each node assigned its own community), and modified to only consider the move of a node to its right-side neighbouring community. As we have shown that surprise-based segmentations already achieve higher similarity metrics (including modularity, which is the objective used in the Louvain method) than fixed or random segmentations, we believe this is a good initialisation as it means that our algorithm will, at worst, achieve the same modularity. While the Louvain method is considered to be an efficient way to converge to _local_ optima when iterated, our own modifications and lack of iterations mean we cannot claim such behaviour but rather suggest that we are likely to see some improvements in our metrics, as our results have confirmed.  We updated the wording used in the main text to clarify these points, as well as discussed this further in the Appendix.
>
>     * **References that do not appear in the manuscript:**
>         * (Blondel et al, 2008), "Fast unfolding of communities in large networks"
>
>     * **Summary of changes made to the paper:**
>         1. Changed "iteratively" to "sequentially" in **Section 3.3**.
>         2. Added a section to the **Appendix** with this discussion.
>         3. Made minor adjustments to the complexity analysis for **Algorithm 1** following internal speculation as to its clarity.

---

> ### Author Response · Authors · 2024-11-19
> **(Answer 2/4)**
>
> * **Reviewer:** _Does the issue of needing to query too many matching events still exist when the context window is too long?_
>
>     **Authors:** This question fits in well with our own motivations and results as the issue of diluted (a.k.a. distracted) attention in full-context transformers (Tworkowski et al, 2023; Ye et al, 2024; Liu et al, 2024), when applied to long-context tasks, is a key motivator for retrieval methods. We believe that our method addresses this issue, both by leveraging k-NNs for top-k retrieval of only the most relevant memory units, and improving the cohesion of keys within such units in order to increase the overall relevance of its member keys. We have mentioned diluted attention, and related works that address it, in section 2.1, and believe that it is responsible for the improvements in $\infty$-Bench's retrieval tasks over full-context and RAG methods. Hence, while the issue of "too many matching events" may still be considered to be present in the k-NN retrieval step, the performance improvements observed from only considering the top-k events and improving on the accuracy of retrieval suggests that our approach can at least reduce this issue. Not to mention the greatly reduced computational complexity of computing a much smaller attention matrix.
>     * **References that do not appear in the manuscript:**
>         * (Ye et al, 2024), "Differential Transformer"
>
> * **Reviewer:** _If we use the Surprise-based method to segment the context into a RAG format, will the performance remain the same?_
>
>     **Authors:** Thank you for this interesting suggestion. To be able to answer your question, we explored this exact approach by implementing a RAG variant that uses our surprise-based segmentation instead of fixed-length chunks and we have now included this information in Supplementary Table 9 and Section 4.4. Specifically, we used the same surprise threshold mechanism from EM-LLM to segment the context before encoding it into the RAG vector database. However, our experiments on LongBench showed that this approach actually performed worse than standard fixed-length chunking in RAG (see  Table 9), with the same context window size. We believe there are several key reasons for this:
>
>     1. Our analysis (see Supplementary Figure 5) shows that $10-60$% of the events recalled by each layer of the LLM are unique to that layer at each query step. This demonstrates that different layers benefit from accessing different parts of the context at different times. In contrast, RAG's single retrieval step forces all layers to work with the same retrieved context, limiting the model's ability to access task-relevant information in a layer-specific way.
>
>     2. Many tasks in our benchmarks (like summarization or multi-hop reasoning) require retrieving different pieces of information at different stages of generation. EM-LLM's layer-wise retrieval naturally supports this by allowing the model to access different context as needed throughout the generation process. RAG's one-time retrieval at the start of generation cannot adapt to these changing information needs.
>
>     3. Since RAG performs only a single retrieval step, its performance heavily depends on getting as much potentially relevant information into the context window as possible. Fixed-length chunks ensure consistently large segments of context, while surprise-based segments can be quite small when detecting frequent meaningful boundaries. This makes fixed-length chunking more effective for RAG's retrieval strategy, even though surprise-based segmentation works better with EM-LLM's more flexible layer-wise retrieval mechanism.
>
>     This finding actually reinforces why EM-LLM's approach of combining surprise-based segmentation with layer-wise retrieval is important - it allows the model to dynamically access different parts of the context in ways that RAG's architecture cannot support.
>
>
>     * **Summary of changes made to the paper:**
>         1. Added results to Appendix **Table 9**.
>         2. Added details describing this new experiment in **Appendix A.2**.
>         3. Added a reference to this new experiment in **Section 4.1**.
>
>
> * **Reviewer:** _How does the method handle dynamic changes in context relevance over time?_
>
>     **Authors:**  After discussing this with our whole team, we have found it difficult to agree on a common interpretation for the reviewers' question. If the reviewer could please provide more details, or an example to illustrate the point touched on in this particular question, we will be happy to address it.

---

> ### Author Response · Authors · 2024-11-19
> **(Answer 3/4)**
>
> ### Responses to points in weaknesses (not addressed in questions):
>
> * **Reviewer:** _The paper claims surprise-based segmentation is superior to fixed-length approaches (like InfLLM), but lacks theoretical justification for why this should be effective. While Figure 4 shows some empirical correlation with human segmentation, there's no analysis of why this leads to better LLM performance. The authors should provide a theoretical analysis showing why this metric captures semantically meaningful boundaries better than alternatives._
>
>     **Authors:** Thank you for your insightful feedback. Whilst we recognize that our initial submission lacked a thorough theoretical justification for the effectiveness of surprise-based segmentation, fixed-length segmentation methods, like InfLLM, are arbitrary and lack support from linguistic or cognitive theories. Fixed length approaches are unable to guarantee segmentation of text into blocks that are contextually cohesive and distinct from each other. As an example, within a text that covers multiple distinct topics, identifying where one topic ends and another begins is challenging with fixed-window segmentation because it imposes artificial boundaries that may split coherent topics or merge distinct ones. Our surprise-based segmentation approach is inspired by cognitive science and aligns with how humans implicitly segment experiences by naturally identifying boundaries where there are shifts in meaning, context, or topic, and is supported empirically by our own results as well as those in neuroscientific literature (see sections 2.2 and 5).
>     We have also shown that surprise-based segmentations have higher similarity metrics than their fixed-length counterparts, and are therefore better at clustering similar keys together. As detailed in section 3.3, the utility of elements within an event, during memory recall for attention, depends on their likelihood of being utilised by the current query. Hence, by improving the similarity of keys within a memory unit, we also make the representative tokens more representative of the constituent keys. Therefore, we are more likely to increase the amount of useful information contained within the retrieved units, while also decreasing the amount of values which would have had low attention scores and potentially only contribute noise.
>     While most of our justifications rely on empirical results, we hope that this is adequate in addressing your point. If you still feel that some of these points are not clear enough in the paper, we will be happy to make changes to this end. As for a more "theoretical" justification for the performance increase, such as mathematical proofs, we believe this is not trivial to come by in a deep-learning setting where most problems are non-convex, and most research is therefore inherently empirical.
>
> * **Reviewer:** _Although Table 1 shows some improvements over InfLLM, the improvements on most tasks may not be statistically significant._
>
>     **Authors:** While the vast majority of our results show a statistically significant improvement on InfLLM at the benchmark level ($p < 0.05$ using a two-tailed z-test, in all LLMs except Phi-3.5), it is true that this isn't the case in the majority of individual tasks. However, given the consistency and frequency of improvements across a large number of such tasks, along with the benchmark-level significance of such improvements, we consider the lower task-level significance to be largely due to the sample size of individual tasks rather than chance, and believe it is still reasonable and justified to claim an overall improvement on InfLLM. Moreover, including individual task results supports transparency and allows for future works to make more granular comparisons and use of such results. Of course, statistical significance testing is a very important part of empirical research so we made the following changes with the aim of being more transparent on this point:
>
>     * **Summary of changes made to the paper:**
>         1. Updated **Appendix A.1** to discuss the significance of results.
>         2. Updated the caption of **Table 1** to point to this addition in **Appendix**.

---

> ### Author Response · Authors · 2024-11-19
> **(Answer 4/4)**
>
> * **Reviewer:** _The human correlation study in Section 4.2 relies on 3 short podcasts, which is insufficient to make broad claims about alignment with human event segmentation. The authors should expand the section to a larger, more diverse dataset of human annotations._
>
>     **Authors:** We appreciate the reviewer's concern about dataset size, as large-scale evaluation is indeed crucial in machine learning. However, we would like to clarify several important points about the human correlation study:
>
>     1. The datasets used in Section 4.2 represent, to the best of our knowledge, some of the most comprehensive text-based human event segmentation annotations available in the cognitive psychology literature. These are not simply short podcasts, but rather carefully curated datasets with rich annotations from 200 human participants, which have been pivotal in recent psychological studies of human event cognition (Lositsky et al., 2016, Michelmann et al., 2021, Kumar et al., 2023). The length and complexity of these narratives (7-30 minutes each) actually make them substantially more extensive than typical stimuli used in human cognitive studies. However, should the reviewer be aware of any other large datasets with robust annotations describing human event cognition, we would be happy to expand our analysis to include it.
>
>     2. The statistical significance and consistency of our results across these datasets, particularly the striking difference in Wasserstein distance between our methods and random/fixed segmentation, suggests that the signal we observe is robust despite the dataset size. This is further supported by the fact that both surprise-based segmentation and refinement show consistent improvements across base LLM models and podcasts (see Figures 6-8).
>
>     3. We fully acknowledge the machine learning community's emphasis on large-scale validation. This is precisely why our broader evaluation of EM-LLM's effectiveness extends well beyond these human correlation studies in two large benchmarks. In addition, to better support our results on online clustering beyond the human-annotated datasets, we conducted extensive experiments on PG-19, a standard benchmark for long-context language models, to validate our approach's scalability and effectiveness in a large-scale setting (see Table 2).
>
> * **Reviewer:**
>     * _The two-stage memory retrieval process (Section 3.4) introduces ks and kc parameters for similarity and contiguity buffers, but it does not justify the choice of values or analyze their trade-offs._
>     * _Section 4.4 discusses the impact of contiguity but does not properly ablate different buffer sizes or analyze the trade-off between similarity and contiguity buffers._
>
>     **Authors: (combined answer)** This is currently visualised in Appendix Figure 11, and we regretably do not have space for it in the main text. However, we agree that this is an important point to touch on and to this end we are adding more details on this, as well as other ablations, and the resulting choice of parameters to the Appendix. We also mention this section briefly in the main text. Specifically:
>
>     * **Summary of changes made to the paper:**
>         1. Pointed to **Appendix D** in caption for **Table 1**.
>         2. Added a sub-section to **Appendix D** providing further details for our initial ablations and parameter tuning.

---

> > ### Author Response · Authors · 2024-11-29
> >
> > Dear Reviewer RkzP,
> >
> > We wanted to kindly remind you that the discussion period ends Monday, December 2nd AoE. We have provided detailed responses to your initial concerns and would greatly value your feedback on whether our clarifications have addressed your questions satisfactorily, or whether there are more changes you would like us to consider. Given the approaching weekend, we understand time might be limited, but any engagement before the deadline would be much appreciated.
> >
> > Best regards,
> > Authors

---

### Official Review · Reviewer_pMUj · 2024-11-03

**Soundness:** 4
**Presentation:** 4
**Contribution:** 4
**Rating:** 8
**Confidence:** 5

**Summary:**

The paper introduces EM-LLM, a novel architecture that enhances the memory capabilities of LLMs by integrating design elements of human episodic memory. EM-LLM employs surprise-based event segmentation, similarity-based, and contiguity-based retrieval to enable LLMs to manage long contexts with computational efficiency. Tokens are organized into episodic events by dynamically detecting boundaries using a surprise metric and further refines these boundaries with graph-theoretic measures. This episodic structure allows EM-LLM to recall information from both the current context and prior temporally contiguous events. Experiments on long-context benchmarks show that EM-LLM outperforms InfLLM and RAG. Additionally, the segmentation results are aligned with human-perceived event structures, suggesting EM-LLM as a computational model of human episodic memory.

**Strengths:**

Originality: EM-LLM’s incorporation of episodic memory features, particularly surprise-based segmentation, similarity-based, and contiguity-based retrieval, is a novel approach within the domain of LLMs. Using LLM surprise to model event cognition and to approximate episodic memory is original for cognitive modeling. This work not only extends the long-context capability of LLMs but also bridges machine learning and cognitive science, presenting a unique computational framework for studying memory and event cognition.
Quality: The paper demonstrates EM-LLM’s performance on recognized long-context benchmarks. The experiments highlight the model’s improvements in accuracy and efficiency over competing methods, including InfLLM and RAG.
Clarity: The paper clearly presents the concept of episodic memory in LLMs, with a logical breakdown of memory segmentation, refinement, and retrieval mechanisms. The figures generally illustrate the processes well.
Significance: EM-LLM has potential significance in advancing LLM capabilities for long-context tasks, a critical area of LLM development. Beyond practical applications, EM-LLM contributes to interdisciplinary research, providing insights into cognitive science by modeling elements of human episodic memory within an LLM framework.

**Weaknesses:**

- In Table 1, the performances of S, SM, and SM+C methods are difficult to directly compare since different base LLMs are used in each row. I know the comparison is (partially) in Fig. 4-7 but they are not mentioned in the main text. It would be helpful if explicitly mentioned/referenced in the main text.
- Although InfLLM is a primary benchmark comparison, it is absent from Figure 1. Including InfLLM in Figure 1 would provide a more complete comparison.
- It’s unclear from the current paper how the number of retrieved events and the number of evicted tokens influence performance. Any plot showing the model’s performance as a function of these parameters would be helpful.
- In Formula (1), notations (\mu_{t-\tau:t}) should be corrected, keeping consistent across the context.
- Certain figures would benefit from clarification, as detailed in questions.

**Questions:**

- Can the authors discuss the feasibility and potential advantages of end-to-end training of networks for event segmentation, refinement, and retrieval, compared to these hand-designed algorithms? As a comparison, these algorithms are implemented by neural circuits in the brain.
- Can the authors discuss how might other features of human episodic memory, such as the hierarchical organization of events (human episodic memory structures can have multi-level segmentation of experiences), benefit EM-LLM’s design in terms of performance?
- Clarifications for Figures:
In Figure 1, does the data reflect results from S, SM, or SM+C?
In Figure 2, the memory units seem to have variable sizes, different from the fixed size in InfLLM? Additionally, how are the bars normalized?
In Figure 4A, S appears closer to human segmentation than SM or SC, contrasting with Figure 4B.
In Figure 4, what does SC stand for? Does "C" refer to the continuity buffer or represent something else?

---

> ### Author Response · Authors · 2024-11-19
> **(Answer 1/4)**
>
> We sincerely appreciate reviewer pMUj's thorough analysis, positive comments and constructive feedback. We have revised our manuscript accordingly and provided detailed responses to each point below.
>
>
> ## Weaknesses:
>
> * **Reviewer:** _In Table 1, the performances of S, SM, and SM+C methods are difficult to directly compare since different base LLMs are used in each row. I know the comparison is (partially) in Fig. 4-7 but they are not mentioned in the main text. It would be helpful if explicitly mentioned/referenced in the main text._
>
>     **Authors:** Thank you for the suggestion. We agree with your observation and have clarified the main text by mentioning such figures (**Section 4.1**, paragraph **2**, where Table 1 is first mentioned, as well as pointing to it in the caption for **Table 1**).
>
> * **Reviewer:** _Although InfLLM is a primary benchmark comparison, it is absent from Figure 1. Including InfLLM in Figure 1 would provide a more complete comparison._
>
>     **Authors:** We understand your motivation for such a figure, but, having considered such a version ourselves, we found that a fourth addition is incompatible with our preferred superimposed format for the figure, making it less clear. Moreover, we are of the opinion that such a comparison is not directly relevant to the content of Figure 1, for which the main focus is a comparison of our retrieval method with full-context and RAG. Instead, a much more detailed comparison of our approach with InfLLM is available in **Table 1** in the main text, as well as Appendix **Tables 3-7**. Of course, should you still feel that such an addition is crucial to this figure, we are open to reconsider.
>
> * **Reviewer:** _It’s unclear from the current paper how the number of retrieved events and the number of evicted tokens influence performance. Any plot showing the model’s performance as a function of these parameters would be helpful._
>
>     **Authors:**  We do agree that an ablation study on the size of the retrieved buffer, and local tokens, as a function of context length, which is proportional to the number of evicted tokens for a fixed-length retrieved buffer, would be a valuable addition to the paper. To this end, we plan to add a figure to the revised version which express our current results as a function of the context length of the evaluated examples. In our current experiments, we have chosen our buffer sizes to align with related works (namely InfLLM) in order to make direct performance comparisons. Such values also keep buffer sizes shorter than the average number of tokens in the evaluated benchmarks to ensure an appropriate use of retrieval. In order to further explore variations in these parameters, we also:
>     1. Ran a small ablation study varying the size of the retrieved buffer for summarization tasks on LongBench.
>     2. Added a table and figure, as well as a discussion, to **Appendix D** which expresses the ablation results as a function of the context length of the evaluated examples.
>
>     However, for the following reasons, we believe this provides limited information on such parameters. For such a study, we have to choose a base LLM trained with a relatively large context window, such as Mistral or LLaMa 3.1 which support context lengths of up to 32K and 128K respectively, in order to ensure that the underlying model can support an adequate range of buffer sizes. As LongBench may be considered relatively short compared to these context windows (average number of tokens per example: 18K$\pm$1.8K with Mistral's tokeniser), $\infty$-Bench would be more appropriate (average number of tokens per example: $> 100$K). Unfortunately, evaluating larger buffer sizes (hence larger attention matrices) on the already-expensive $\infty$-Bench benchmark would be a very demanding ablation given our limited hardware resources, and hence we have left it for future work and mentioned it as such in the Discussion.
>
> * **Reviewer:** _In Formula (1), notations_ $\mu_{t-\tau:t}$ _should be corrected, keeping consistent across the context._
>
>     **Authors:** Thank you for catching this mistake, we have now fixed **Eq. 1** in the revised version.
>
> ## Questions:
>
> * **Reviewer:** _Can the authors discuss the feasibility and potential advantages of end-to-end training of networks for event segmentation, refinement, and retrieval, compared to these hand-designed algorithms? As a comparison, these algorithms are implemented by neural circuits in the brain_
>
>     **Authors:** This is a great question. We present a brief discussion below. Due to space limitations, we haven't included this in the main paper, but we welcome your thoughts on whether it should be added to the supplementary material.
>
> (continues to the next official comment..)

---

> > ### Author Response · Authors · 2024-11-19
> > **(Answer 2/4)**
> >
> > * **Event segmentation:** Differentiable event segmentation models have already demonstrated the feasibility of learning a temporal structure from continuous experience. Models like SEM (Franklin et al., 2023) show how neural networks can combine with probabilistic inference to capture human-like event segmentation, while approaches like the DLH (Zakharov et al., 2022b) demonstrate that neural architectures can learn to identify hierarchical temporal boundaries through differentiable clustering and amortised variational inference. For instance, using the VaDE trick in (Jiang et al., 2017). These approaches offer powerful advantages in terms of learned representations and flexibility, potentially capturing the complex hierarchical event structure of real environments and adapting to different domains. Particularly compelling advantages include the ability to perform layer-wise or attention-head-wise segmentation and the potential emergence of nested timescale structures, as demonstrated in (Zakharov et al., 2022a;b), mirroring how the brain processes events at multiple temporal scales (Baldassano et al., 2017). While such end-to-end training is theoretically appealing and mirrors how neural circuits might learn temporal structure, our method takes a more pragmatic approach by leveraging the pre-trained capabilities of LLMs. By using Bayesian surprise computed directly from model outputs to detect event boundaries, we achieve efficient segmentation without requiring complex architectural modifications or additional training, while still aligning with cognitive theories about prediction errors in event perception (Zacks et al., 2007).
> >
> > * **Retrieval:** The development of neural architectures for memory retrieval has evolved from classical Hopfield networks (Hopfield, 1982) through several key innovations. Early Hopfield networks demonstrated how content-addressable memory could emerge from simple neural circuits, paralleling biological memory systems. This was significantly advanced by Neural Turing Machines (Graves et al., 2014) and their successor, the Differentiable Neural Computer (Graves et al., 2016), which introduced differentiable memory access mechanisms. Modern Hopfield networks (Ramsauer et al., 2020) further revolutionized our understanding by establishing a theoretical connection between transformer attention and associative memory, showing how these systems can store and retrieve exponentially many patterns while maintaining stable dynamics. Such end-to-end approaches could particularly benefit the quality of memory representations, as they could learn optimal projections for generating representative keys for memory blocks, potentially capturing universal contextual patterns more effectively than our current approach. While end-to-end training of memory systems is feasible, as demonstrated by models like MERLIN (Wayne et al., 2018), such approaches often face challenges with credit assignment over long sequences and require complex architectural modifications. Our KNN-based approach leveraging the KV cache offers a pragmatic middle ground: it harnesses the rich semantic representations already present in transformer models while maintaining the computational benefits of nearest-neighbour retrieval. This aligns with both biological intuitions about pattern matching in the hippocampus (O'Reilly and Norman, 2002) and the theoretical foundations of modern Hopfield networks, where similarity-based attention serves as a form of associative memory. By operating on pre-trained representations, our method sidesteps the training complexities of fully differentiable memory while preserving the benefits of content-based retrieval.
> >
> > * **Refinement:** The refinement of event boundaries could also theoretically be learned end-to-end, similar to how attention pruning mechanisms (Ying et al., 2019) learn to identify optimal subgraphs in graph neural networks, or how hierarchical clustering can be made differentiable (Ying et al., 2018). Our graph modularity approach provides a computationally efficient alternative that optimizes for coherence within segments while respecting the initial surprise-based boundaries. While our method is primarily motivated by computational considerations, it parallels how memory consolidation might strengthen associations between related elements within an event while weakening cross-event associations (Preston and Eichenbaum, 2013). The high modularity of our surprise-based segmentation, even before refinement, suggests that prediction errors naturally tend to occur at boundaries between coherent event structures.
> >
> > (continues to the next official comment..)

---

> ### Author Response · Authors · 2024-11-19
> **(Answer 3/4)**
>
> * **Reviewer:** _Can the authors discuss how might other features of human episodic memory, such as the hierarchical organization of events (human episodic memory structures can have multi-level segmentation of experiences), benefit EM-LLM’s design in terms of performance?_
>
>     **Authors:** Thank you for this question. We briefly touched on this in the Discussion as well as our previous answer here. Human episodic memory exhibits several key features that could enhance EM-LLM's capabilities:
>
>     * **Hierarchical organisation:** A hierarchical structure in memory can provide multiple advantages such as improved retrieval, more disentangled latent embeddings, longer future predictions (Saxena et al., 2021; Zakharov et al., 2022b), better planning (Hafner et al., 2022) and higher agreement with neural processes in the brain (Baldassano et al., 2017). In our model, hierarchical organisation of episodic memories based on the existing hierarchy of embeddings in the LLM layers could be implemented by extending our segmentation processes to operate at each layer of the Transformer independently. This could be achieved either through a differentiable approach or a layer-specific surprise metric. Interestingly, our current k-NN retrieval approach already implicitly leverages hierarchical structure through its underlying approximate nearest neighbour algorithms, which typically employ tree-based structures (Johnson et al., 2019) to efficiently partition the embedding space.
>
>     * **Memory consolidation:** The brain's process for memory consolidation is crucial for continual learning, an ability that remains largely unsolved in current LLMs. Implementing consolidation mechanisms in EM-LLM could help address catastrophic forgetting while enabling more efficient integration of new information with existing knowledge.
>
>     * **Mental time travel:** The ability to employ the same retrieval mechanism for imagining future events as for recalling past experiences is a key feature of episodic memory that could significantly enhance LLMs' planning and reasoning capabilities. By leveraging its event-based structure to simulate potential future scenarios or recall past experiences in novel contexts, this mechanism could provide a powerful solution for planning and reasoning, which are currently important challenges in large generative models.
>
> * **Reviewer:** _Clarifications for Figures: In Figure 1, does the data reflect results from S, SM, or SM+C? In Figure 2, the memory units seem to have variable sizes, different from the fixed size in InfLLM? Additionally, how are the bars normalized? In Figure 4A, S appears closer to human segmentation than SM or SC, contrasting with Figure 4B. In Figure 4, what does SC stand for? Does "C" refer to the continuity buffer or represent something else?_
>
>     **Authors:** Thank you for pointing out any miscommunication in these figures, we have now updated their descriptions. In particular:
>
>     * *Figure 1*: The data here reflects the results from surprise-based (**S**) segmentation, as denoted by "$EM-LLM_S$". We followed the same notation as in the other tables (notably **Table 1**), and have updated the caption of **Figure 1** to clarify this.
>
>     * *Figure 2*: We have used this figure to illustrate that k-NN retrieval of past tokens for softmax attention can be seen as a form of hierarchical attention. Please note that this illustration is relevant for any form of k-NN retrieval including both InfLLM and EM-LLM. However, we understand that the coincidental placement of this figure immediately after mentioning InfLLM's fixed-size segmentations in section 2.1 made this confusing, as the figure does not show fixed-sized units. We have now clarified this by re-wording the reference to **Figure 2** in this section. As for the normalisation, this is simply the softmax of the similarity scores shown, as it is describing softmax attention. We also updated the legend in **Figure 2** to clarify this.
>
>     * *Figure 4*: In this figure, no contiguity is shown. As such, **SC** follows the format of **SM**, introduced in the caption for **Table 1**, and shows surprise-based boundary refinement with _conductance_ as the similarity metric. We understand this was not clear and thank you for pointing it out. We have updated the caption of **Figure 4** to clarify this. Finally, **Figure 4A** differs from **4B** in that it shows the average similarity metrics for events due to various segmentation methods, while **4B** shows the distance in actual event positions from the human data. With this in mind, our interpretation of such a discrepancy is that, while two segmentation methods may achieve similar similarity metrics, it does not necessarily mean that their event boundaries are close to each other. Hence, in this specific case, it would appear that SM and SC are both closer to human-perceived event boundaries than S, while still achieving better similarity metrics.

---

> > ### Author Response · Authors · 2024-11-19
> > **(Answer 4/4) - References**
> >
> > **References that do not appear in the manuscript:**
> > * (Jiang et al., 2017) "Variational deep embedding: an unsupervised and generative approach to clustering", IJCAI'17.
> > * (Franklin et al., 2020) "Structured Event Memory: A neuro-symbolic model of event cognition." _Psychological review_ 127.3 (2020): 327.
> > * (Hopfield, 1982). "Neural networks and physical systems with emergent collective computational abilities", PNAS 79.8 (1982): 2554-2558.
> > * (Graves et al., 2014) "Neural Turing Machines", arXiv:1410.5401.
> > * (Graves et al., 2016) "Hybrid computing using a neural network with dynamic external memory", _Nature_ 538.7626 (2016): 471-476.
> > * (O'Reilly and Norman, 2002). Hippocampal and neocortical contributions to memory: Advances in the complementary learning systems framework. _Trends in Cognitive Sciences_ 6.12 (2002): 505-510.
> > * (Ramsauer at al., 2020). Hopfield Networks is All You Need. _arXiv:2008.02217_
> > * (Wayne et al., 2018). Unsupervised Predictive Memory in a Goal-Directed Agent. arXiv:1803.10760.
> > * (Ying at al., 2019). GNNExplainer: Generating explanations for graph neural networks. NeurIPS, 2019.
> > * (Ying at al., 2018). Hierarchical graph representation learning with differentiable pooling. NeurIPS, 2018.
> > * (Preston and Eichenbaum 2013). Interplay of hippocampus and prefrontal cortex in memory. _Current biology_ 23.17 (2013): R764-R773.
> > * (Saxena et al., 2021) "Clockwork variational autoencoders."_Advances in Neural Information Processing Systems_ 34 (2021): 29246-29257.
> > * (Hafner et al., 2022) "Deep hierarchical planning from pixels."_Advances in Neural Information Processing Systems_ 35 (2022): 26091-26104.
> > * (Johnson et al., 2019) "Billion-scale similarity search with GPUs."_IEEE Transactions on Big Data_ 7.3 (2019): 535-547.

---

> ### Comment · Reviewer_pMUj · 2024-11-26
>
> I appreciate the authors' thoughtful clarifications. The current Figure 1 is clear. Thank you for the additional ablation study and detailed discussion, which all could be added to the supplementary material.
>
> Event boundary segmentation (Zacks et al., 2007, Baldwin & Kosie, 2020; Baldwin et al., 2008; Schapiro et al., 2013) and asymmetric contiguity biases (Kahana, 1996; Murdock, 1962; Murdock & Okada, 1970) are among the most significant well-established properties of biological episodic memory, extensively studied in the cognitive science literature. The authors' correct identification of these crucial elements and their integration into the models to improve performance over competing approaches like InfLLM is both impressive and insightful. By incorporating these principles, the mechanisms of LLMs are rendered more human-like, enhancing their alignment with biological cognitive processes.
>
> Your work stands out as an exemplar of how cognitive science and neuroscience-inspired designs can improve LLM models. Moreover, the comparison with human event segmentation in a human-annotated audio dataset provides a unique and valuable bridge, leveraging LLM representations to deepen our understanding of human episodic memory. This innovative approach has significant potential to inspire further interdisciplinary exploration at the intersection of AI, cognitive science, and neuroscience.

---

### Official Review · Reviewer_SyTr · 2024-11-03

**Soundness:** 2
**Presentation:** 3
**Contribution:** 2
**Rating:** 5
**Confidence:** 4

**Summary:**

In this work, the authors proposed a framework, termed EM-LLM, for LLMs to store previously seen sequences of tokens as memory and then utilize the stored information to improve the performance of newly encountered problems. Specifically, inspired by human episodic memory, the authors suggested a method based on Bayesian surprise and graph-theoretic refinement to parse the sequences into events. Later, these store events were retrieved depending on content similarity and temporal contiguity. Experiments were conducted to demonstrate the performance improvements. In addition, a comparison was conducted to show that the parsing of events by the proposed method was similar to the results of humans, suggesting a potential link to the underlying mechanism in the brain.

**Strengths:**

Memory is a key function in biological brains that helps organize experiences and use them to guide future behaviours.  The lack of memory function in the  LLM is a major aspect that needs to be addressed. The authors have proposed a framework inspired by the biological memory system for LLMs, which is desirable and may stimulate further investigations in this direction.  Overall, the paper was clearly written, showing comprehensive experimental results.

**Weaknesses:**

"Human-like Episodic Memory" is an overstatement. First, episodic memory combines multimodal information into a coherent recollection of experiences, with key aspects including when, where, what, how and with who a event happened. To store a sequence of tokens does not necessarily reflect such a memory. Second, parsing of events in human memory depends heavily on the content, e.g., change of location or context. The Bayesian surprise measure may be a useful proxy to approximate such separation, but it does not by itself suggest a common mechanism, nor support the claim of "human-like". Third, the retrival rule of similarity is generic for memory, not specific for EM.

Another concern is that the memory-based frawmork has been previously proposed (cited as Xiao et al, 2024a in the current paper). The present work is an refinement of the framework, which somewhat limits its noverty.

**Questions:**

1. I would suggest the authors to better explain the differrence in comparison with Xiao et al. 2024a, and explain why it is an important conceptual advance, instead of an incremental improvement.
2. To better justify the claim that the mechanism is EM-related with further analyses.

---

> ### Author Response · Authors · 2024-11-19
> **(Answer 1/3)**
>
> We sincerely thank the reviewer for their thorough analysis. We particularly appreciate their concern about potential overstatements in our terminology. We want to emphasize that our work claims similarities to, not equivalence with, human episodic memory - a distinction we should have made more explicit. While we provide detailed empirical evidence supporting these similarity claims in the responses below, we are also open to adjusting terminology (e.g., from "human-like" to "human-inspired") if our justification does not fully address the reviewer's concerns.
>
> ## Question 1:
>
> We thank the reviewer for this suggestion. We made amendments to our paper to clarify this very important point, which we summarise here.
>
> The reviewer is correct to point out that the most similar architecture to our work is the model proposed by (Xiao et al. 2024a), namely InfLLM, which extended the KV lookup method introduced in (Han et al., 2023) for groups of tokens, by segmenting the context window into equally-sized blocks. Building on these previous methods, in this work we have made *three* novel architectural contributions for LLMs, for which we show their importance both conceptually and with empirical results:
>
> 1. **Dynamic surprise-based segmentation.** We introduce the first method for dynamic segmentation of KV cache into blocks. Our method is also the first that manipulates the KV cache based on insights from cognitive science, using an intrinsic measure to LLMs. The only study in the literature that concerns Transformers and suggests a connection with surprise, was a psychology study (Kumar et al., 2023) that showed a connection between a similar measure of surprise and human event segmentation without, however, proposing or attempting to use this insight to alter the Transformer architecture in any way. We show empirically using multiple LLMs that this low-cost and simple-to-implement method is able to group relevant pairs of keys and values (KV) together (relevance measured as key similarity) with much higher accuracy than fixed segmentation, which is the only alternative proposed approach (See Table 2 for key similarity comparisons). We also show that this method results in increased LLM performance, especially in retrieval tasks ($16.6$% average increase over InfLLM) and multi-document QA tasks ($6.4$% average increase over InfLLM) across all the LLMs we tried (See the "S" column in the tables of Appendix A.1).
>
> 2. **Graph-based refinement.** We were the first to introduce an algorithm to refine the temporal borders of events in the context window of LLMs using graph theory. We relied on the insight that tokens are more useful when recalled together if the variance between their keys is low, as they are retrieved using a single query at a time. This method can also stand by itself as a dynamic segmentation approach of KV cache, more computationally heavy than surprise-based segmentation but achieving a competitive accuracy in grouping relevant (KV) together (see again Table 2), with the added benefit that it can be used in each attention head independently, without relying on the LLM output.
>
> 3. **Contiguity buffer**. We were the first to introduce a method to maintain a dedicated decaying buffer in the context window of LLMs that maintains the KV cache of temporally contiguous tokens to the context window of the LLM for a certain amount of time. For this, we relied on the recent insight that self-attention heads responsible for in-context learning are shown to consecutively attend to contiguous groups, similarly to human studies (Ji-An et al., 2024). We show that this algorithm can also be combined with methods (1.) and (2.) and results in further increases in the overall LLM performance. Notably, the average increase in retrieval tasks over InfLLM jumps to $19.4$%, and for multi-document QA tasks to $9.2$% across all the LLMs we tried (See the "SM+C" column in the tables of Appendix A.1).
>
> ### Summary of changes made to the paper:
> 1. We have updated the last two paragraphs of the **Introduction** to clarify our contributions.
> 2. We have introduced the **Appendix E.2** section (referred to in the **Introduction**) to further clarify this by reiterating the points in this response.

---

> > ### Comment · Reviewer_SyTr · 2024-11-26
> >
> > Thank you for the clarifications; they are helpful. However, I still perceive the conceptual improvements as somewhat incremental, as the current work appears to be a refinement of the previous study (InfLLM).

---

> > > ### Author Response · Authors · 2024-11-26
> > >
> > > Thank you for your response. Regarding our contribution, we consider it sufficient for this ongoing field of work (3 distinct architectural novelties, current SOTA performance in different benchmarks, and context lengths far beyond anything in the literature of Transformers), especially considering the extensive experimentation, analysis, and resulting insights.
> > >
> > > Could the reviewer please elaborate on what they would like us to include, for our contribution to not be considered incremental?

---

> > > > ### Comment · Reviewer_SyTr · 2024-11-27
> > > >
> > > > I am not questioning about the validity of the work. The proposed approach worked well, brining solid improvements. What I said was that the conceptual framework is similar to InfLLM, with refined process of memory formation, consolidation and retrieval. And that's ok.

---

> ### Author Response · Authors · 2024-11-19
> **(Answer 2/3)**
>
> ## Question 2:
>
> We thank the reviewer for their thoughtful critique regarding our claims of human-like episodic memory. We agree that such claims require careful justification, and we appreciate the opportunity to clarify how our model addresses each aspect of episodic memory raised in the review. Please consider the below, which is also amended in the latest version of the manuscript.
> ### 1. Foundation: Transformers and Episodic Memory
> We completely agree that all the points mentioned regarding what features a human-like model of episodic memory (EM) should incorporate are important and valid. As the reviewer states, EM indeed combines multiple pieces of information into a coherent recollection of experiences. In Transformers, this integration naturally occurs through latent embeddings - recollections of concepts inferred from inputs and encoded in the embedding space, recalled via the SoftMax self-attention mechanism. Recent work has shown this enables human-like behaviour in short-context memory recall tasks (Ji-An et al., 2024).
> However, in long-context tasks, transformers face two fundamental constraints that break this connection to human episodic memory:
> - Computational and memory complexity increases quadratically.
> - Retrieval performance drops due to attention dilution (Tworkowski et al., 2023; Ye et al., 2024).
> ### 2. Our Solution
> Considering the above, the main purpose of any memory architecture claiming human-like EM functionality must be to extend transformers' inherent capabilities beyond these limitations. Our proposed method achieves this to a degree that goes far beyond the computational capabilities of full-context LLMs today (at least 10M tokens using a single GPU). To do this, we introduce a connection to human event cognition, which we claim is precisely what's missing from standard transformers. While they can integrate information within their attention window, they lack the systematic event-based organization that characterizes human EM. Our method achieves this through two key mechanisms:
> #### A. Information Grouping Through Event Segmentation
> For grouping information together, we use Bayesian surprise, which has been repeatedly shown to capture content-dependent event boundaries in human perception. This choice is deeply grounded in both behavioural and neuroimaging evidence showing that surprise (or prediction error) signals in the hippocampus and cortical regions are crucial for episodic memory and event boundary formation (Sinclair et al., 2021; Zacks et al., 2007; 2011; Sherman et al., 2022;
> Mariola et al., 2022; Fountas et al., 2022), including direct empirical evidence with our particular implementation presented in our paper. Our analysis shows that this approach effectively groups together semantically related content: tokens within surprise-based segments have significantly higher key similarity than tokens across segments (see Table 2), indicating natural capture of meaningful content transitions. This aligns with both empirical studies of human event segmentation and the reviewer's point about content-dependent parsing.
>
> Importantly, due to the nature of Transformers, the contents of experience (embeddings) are processed within the temporal limits of these events. This allows us to effectively handle the key aspects of episodic memory that the reviewer identified:
> * **When, where, what, how and with who an event happened:** This information, given to an LLM via text during an episode, forms the basis of most QnA and retrieval tasks in our benchmarks. Our model's significant performance boost compared to baseline InfLLM demonstrates its ability to maintain and retrieve these coherent aspects of experiences together.
> * **Multi-modality:** While our current implementation focuses on text, our approach is fundamentally compatible with multi-modal processing. In Transformer-based models, integrating information across different modalities follows the same principles as integrating information within a single modality. Recent multi-modal models (e.g., Qwen2-VL, Pixtral, LLaMa 3.2) already demonstrate this by integrating different modality encoders into a single embedding space. Our EM-LLM can accommodate these approaches as the KV cache treats all embeddings equally. This point is now added in the discussion.

---

> > ### Comment · Reviewer_SyTr · 2024-11-26
> >
> > I remain unconvinced that the memory mechanisms implemented here accurately represent episodic memory (EM). As I mentioned in my original comments, EM pertains to the recollection of experiences rather than merely a sequence of texts. For instance, remembering that I read a book constitutes EM, while recalling the book's content does not. Therefore, the assertion that storing sentences is equivalent to human-like EM was not well-supported.

---

> > > ### Author Response · Authors · 2024-11-26
> > >
> > > Firstly, we would like to re-iterate that we do not claim _equivalence_ with EM, nor do we wish to do so, as outlined in our initial response. We are still very much open to updating our wording to this end and still hope to get the reviewer's input as to what level of similarity between humans and EM they believe is more appropriate to claim given our methods, analysis, and results. We believe there is merit to our approach regardless of the exact wording used in this comparison, and are more than happy to adapt it in order to move forward with our work.
> > >
> > > Secondly, we agree that the simple recollection of "sequences of text" may not be enough to constitute an EM-inspired mechanism, but we would like to re-iterate that our approach goes beyond simple text sequence recall, unlike typical RAG methods. While event boundaries are defined at the token level, they are stored as groups of **KV pairs** in each layer and head, not as text. These KVs, generated **per layer** and **head**, enable the model to process and focus on different aspects of the sequence, including necessary contextual information, creating a natural hierarchy across layers as is inherent to Transformers (also see our initial response). The **layer-wise retrieval** of these KVs, which varies significantly across layers, further enhances the complexity of recall compared to sequential text inputs. This results in a more sophisticated information retrieval process which, combined with a transformer, has all the tools to allow for the complete recollection of "experiences". Appendix A.2 illustrates the behaviour of layer-wise retrieval and demonstrates that our approach vastly outperforms standard and surprise-based RAG as a result.

---

> > > > ### Comment · Reviewer_SyTr · 2024-11-27
> > > >
> > > > Thanks for the further clarification. Based on the evidence provided so far, it is difficult to determine whether it is more like episodic memory or declarative, non-episodic memory. So, I think it would be helpful to discuss such potential controversy, and maybe also tune down the tone (e.g., “episodic memory inspired” as suggested by the authors).

---

> > > > > ### Author Response · Authors · 2024-11-27
> > > > >
> > > > > Thank you for your quick response and continued feedback. Following your suggestions, we have:
> > > > >
> > > > > 1. Amended the terminology throughout the paper (including **Title**, **Abstract**, **Introduction**, and **Discussion**) to 'episodic memory-inspired' rather than claiming human-likeness.
> > > > > 2. Added an expanded discussion of declarative memory systems, potential controversies and future extensions (**Appendix E.1.4**, **E.4**, and **E.5**), incorporating insights noted by reviewer pMUj regarding the cognitive science foundations of our approach (https://openreview.net/forum?id=BI2int5SAC&noteId=f2FlQ3nxyh).
> > > > >
> > > > > We appreciate your thoughtful engagement with our work and remain open to any additional suggestions you may have.

---

> ### Author Response · Authors · 2024-11-19
> **(Answer 3/3)**
>
> #### B. Human-like Information Retrieval
> The reviewer correctly notes that similarity-based retrieval is not unique to EM. However, our key contribution is integrating this with decaying temporal contiguity, reflecting both human cued recall and free recall patterns (Howard & Kahana, 2002). This combination enables our model to exhibit both temporal contiguity effects and temporal asymmetry - behavioral patterns consistently observed in human EM retrieval. In other words, while similarity-based retrieval is indeed general to many memory systems, our integration with contiguity buffer and temporal organization of memory blocks results in behavioural patterns that both align with key characteristics of human episodic memory and address previously unsolved challenges in long-context LLM tasks.
>
> ### 3. Limitations and Future Work
> We acknowledge that EM-LLM differs from human episodic memory in several important ways (analysed in Appendix E.1). The most important ones we can identify include: (a) the non-parametric nature of our method, (b) the lack of hierarchical event structure, (c) the limited cross-modal integration and, finally, (d) the absence of a memory consolidation framework. However, we believe these limitations represent opportunities for future work rather than fundamental flaws in our approach.
>
> We would appreciate the reviewer's thoughts on whether this analysis adequately addresses their concerns about our model's connection to human episodic memory. Given these limitations and our emphasis on claiming similarities rather than equivalence with human episodic memory, if the reviewer feels our justification above still doesn't adequately support the term "human-like", we would be happy to modify our framing to "human-inspired" episodic memory throughout the paper.
>
> ### Summary of changes made to the paper:
> 1. We have introduced the **Appendix E.1** section to clarify our points here.
> 2. We have added a reference to **Appendix E.1** to the first paragraph of the **Discussion**, which aims to compare EM-LLM to human studies of episodic memory.

---

> > ### Comment · Reviewer_SyTr · 2024-11-26
> >
> > Thanks for the clarifications, which are helpful.

---

### Author Response · Authors · 2024-11-19
**Response to all reviewers**

We sincerely thank all reviewers for their thoughtful feedback and valuable suggestions. We have provided detailed responses to each point raised, and we are currently incorporating these discussions into an updated version of the paper. As noted in our individual responses, some content will be added to the supplementary material due to space constraints. We plan to update the paper PDF later in the discussion period to incorporate any additional feedback that may arise. We believe these revisions will strengthen the paper and clarify important aspects of our work.

---

### Author Response · Authors · 2024-11-22
**Complete paper revision: New results and detailed change log**

Dear Reviewers,

We sincerely thank you for your thoughtful feedback which has helped us significantly strengthen the paper. We have now uploaded a revised PDF that incorporates all discussed changes, with particular focus on the main concerns raised:

1. Empirical validation (due to points raised by **pMUj**, **RkzP**):
    * Included new experiments where:
      * We achieved **100**\% retrieval score in **10M** tokens using a single 32GB NVIDIA GPU and *LLaMA3.1-8B* (see **Fig. 1**). To our knowledge, this is far beyond any length achieved so far with transformer models in the literature.
      * We compared surprise-based segmentation in RAG format and showed it does not beat standard RAG, let alone our approach. We discuss the reasons why.
    * Added statistical significance analysis showing consistent improvements over InfLLM (p < 0.05) in Appendix **A.1**.
    * Expanded ablation studies on parameter sensitivity and buffer sizes in Appendix **D.1**.
2. Novelty and contributions (due to points raised by **SyTr**, **N12r**):
    * Clarified our three key architectural innovations: dynamic surprise-based segmentation, graph-based refinement, and contiguity buffer (see end of **Introduction** and Appendix **E.2**).
    * Enhanced discussion of theoretical foundations linking our approach to cognitive science (see Appendix **E.1**, mentioned also in the **Discussion**).
3. Technical rigour (due to points raised by **RkzP**):
    * Updated complexity analysis for boundary refinement (Appendix **C.1**).
    * Enhanced discussion of k-NN retrieval efficiency (Appendix **C.2**).
    * Expanded parameter tuning methodology (Appendix **D.1**).
4. Clarity improvements:
    * Updated most figures and captions for better interpretation.
    * Strengthened connection between main text and appendix content.
    * Refined terminology where needed for precision.

We have also edited our responses to add details on the specific changes made for each point raised for your convenience. The results continue to demonstrate state-of-the-art performance on long-context tasks while maintaining computational efficiency. We remain fully committed to addressing any additional feedback you may have, whether about terminology, methodology, or experimental validation. We will promptly incorporate your suggestions and upload new revisions as needed.

Best regards,

The Authors

---

### Meta-Review · Area_Chair_2wN3 · 2024-12-20

**Metareview:**

(a) This paper introduces a novel architecture, EM-LLM, designed to enhance the long-context capabilities of Large Language Models (LLMs). The authors draw inspiration from human episodic memory and event cognition to develop a system that can handle theoretically infinite context lengths. The key scientific claims include:
- Dynamic Event Segmentation: EM-LLM segments text into meaningful episodic units using a measure of Bayesian surprise, departing from fixed-length approaches like InfLLM.
- Graph-Based Refinement: The initial segmentation undergoes refinement based on graph-theoretic metrics, optimizing the cohesion within and separation between events for better information retrieval.
- Two-Stage Memory Retrieval: EM-LLM employs a retrieval process that combines similarity-based search with a mechanism mimicking the temporal contiguity effect observed in human recall, enabling more nuanced access to information.
- Good performance on LongBench and ∞-Bench.

(b) Strengths
- The proposed EM-LLM architecture offers a novel approach to long-context LLM processing, combining established techniques (Bayesian surprise, graph-theory, k-NN retrieval).
- The demonstrated ability to process vast text sequences (10 million tokens) with high accuracy showcases the model's scalability and potential for real-world applications where context is crucial.

(c) Weaknesses:
- Limited Human Correlation: While the study attempts to connect EM-LLM to human event perception, the analysis relies on a relatively small dataset (3 podcasts) and lacks deeper exploration of the nuanced relationship between model behavior and human cognitive processes.
- The initial framing as "human-like" episodic memory was considered an overstatement, prompting the authors to revise the terminology to "episodic memory-inspired".
- Some reviewers desired a stronger theoretical grounding for certain design choices, such as why surprise-based segmentation should lead to improved performance compared to fixed-length methods.

(d) The decision is to Accept.
While acknowledging the weaknesses, the novelty of the architecture, the promising empirical results, and the potential for advancing both LLM capabilities and computational models of cognition make this paper a valuable contribution.

**Additional Comments On Reviewer Discussion:**

During the rebuttal period, reviewers raised several concerns, leading to clarifications, additional experiments, and revisions by the authors.
- Concerns about the Novelty and Contributions: Some reviewers found the conceptual improvements incremental, viewing the work as a refinement of InfLLM, and questioned the specific innovations, as many methods seemed previously proposed. The authors acknowledged that some techniques were inspired by prior work but emphasized their unique integration and the significant performance gains achieved.
- Reviewers questioned the justification for claiming the model was "human-like," arguing that episodic memory involved more than text sequences. They suggested alternative terms like "episodic-memory-like" or "episodic-memory-inspired." The authors agreed to adjust the terminology, changing "human-like" to "episodic memory-inspired" throughout the paper.
- Reviewers requested clarification on figures and tables, including details about parameter choices, performance comparisons, and the meaning of certain abbreviations. The authors addressed most of them.

The authors' responsiveness to reviewer feedback, including conducting additional experiments, clarifying their claims, and revising the manuscript, further strengthens the paper's value.

---

### Decision · Program_Chairs · 2025-01-22

Accept (Poster)